# Archetypal Analysis++:
# Rethinking the Initialization Strategy

**Sebastian Mair**                                        *sebastian.mair@it.uu.se*
*Uppsala University, Sweden*

**Jens Sjölund**                                          *jens.sjolund@it.uu.se*
*Uppsala University, Sweden*

**Reviewed on OpenReview:** *https://openreview.net/forum?id=KVUtlM6OHM*

## Abstract

Archetypal analysis is a matrix factorization method with convexity constraints. Due to local minima, a good initialization is essential, but frequently used initialization methods yield either sub-optimal starting points or are prone to get stuck in poor local minima. In this paper, we propose archetypal analysis++ (AA++), a probabilistic initialization strategy for archetypal analysis that sequentially samples points based on their influence on the objective function, similar to $k$-means++. In fact, we argue that $k$-means++ already approximates the proposed initialization method. Furthermore, we suggest to adapt an efficient Monte Carlo approximation of $k$-means++ to AA++. In an extensive empirical evaluation of 15 real-world data sets of varying sizes and dimensionalities and considering two pre-processing strategies, we show that AA++ almost always outperforms all baselines, including the most frequently used ones.

## 1 Introduction

Archetypal analysis (AA) (Cutler & Breiman, 1994) is a matrix factorization method with convexity constraints. The idea is to represent every data point as a convex combination of points, called archetypes, located on the boundary of the data set. Thus, archetypes can be seen as well-separated observations that summarize the most relevant extremes of the data. The convexity constraints also give archetypal analysis a natural interpretation. An example is shown in Figure 1.

Archetypal analysis has been applied, *e.g.*, for single cells gene expression (Thøgersen et al., 2013; Korem et al., 2015), bioinformatics (Hart et al., 2015), apparel design (Vinué et al., 2015), chemical spaces of small organic molecules (Keller et al., 2021), geophysical data (Black et al., 2022), large-scale climate drivers (Hannachi & Trendafilov, 2017; Chapman et al., 2022), and population genetics (Gimbernat-Mayol et al., 2022).

To improve the computation of archetypal analysis, various optimization approaches (Bauckhage & Thurau, 2009; Mørup & Hansen, 2012; Chen et al., 2014; Bauckhage et al., 2015; Abrol & Sharma, 2020) and approximations (Mair et al., 2017; Damle & Sun, 2017; Mei et al., 2018; Mair & Brefeld, 2019; Han et al., 2022) have been proposed. However, the earliest point of attack for obtaining a good solution is the initialization of the archetypes. Surprisingly, this has barely been investigated.

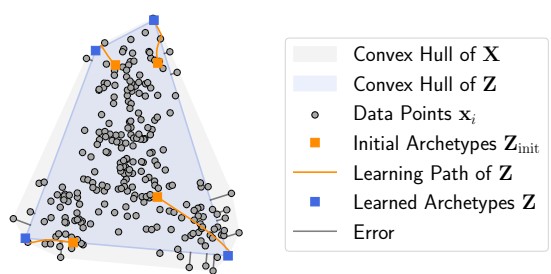

Figure 1: Archetypal analysis in two dimensions with $k = 4$ randomly initialized archetypes $\{\mathbf{z}_1, \ldots, \mathbf{z}_4\}$ shown in orange. The archetypes after optimization are depicted in blue.

In the original paper, Cutler & Breiman (1994) use a random initialization, *i.e.*, choosing points uniformly at random from the data set, which was adopted by many others, *e.g.*, Eugster & Leisch (2011); Seth & Eugster (2015); Hinrich et al. (2016); Hannachi & Trendafilov (2017); Mair et al. (2017); Mei et al. (2018); Krohne et al. (2019); Olsen et al. (2022) to name a few. Furthermore, Cutler & Breiman (1994) state that a careful initialization improves the convergence speed and that archetypes should not be initialized too close to each other.

This idea serves as an argument for using the *FurthestFirst* approach (Gonzalez, 1985; Hochbaum & Shmoys, 1985), yielding a well-separated initialization, *e.g.*, for $k$-means clustering (Lloyd, 1982). Inspired by FurthestFirst, Mørup & Hansen (2010; 2012) propose a modification for archetypal analysis called *FurthestSum*, which focuses on boundary points rather than well-separated points. Here, boundary points refer to points on the boundary of the convex hull of the data. Since then, FurthestSum has established itself as one of the default initialization strategies for archetypal analysis which is used, *e.g.*, by Thøgersen et al. (2013); Hinrich et al. (2016); Mair & Brefeld (2019); Abrol & Sharma (2020); Beck et al. (2022); Black et al. (2022); Chapman et al. (2022); Gimbernat-Mayol et al. (2022).

Despite its popularity, FurthestSum has also been criticized. For example, Suleman (2017) states that FurthestSum is prone to selecting redundant archetypes, primarily when many archetypes are used. Redundant archetypes lie in the convex hull of the already selected archetypes and thus do not lower the overall error. In addition, Krohne et al. (2019) and Olsen et al. (2022) report better results with a random uniform initialization than with FurthestSum. A possible explanation is that FurthestSum's early focus on sub-optimal boundary points risks trapping the optimization of archetypal analysis in poor local minima.

**Contributions.** In this paper, we (i) propose archetypal analysis++ (AA++), an initialization strategy inspired by $k$-means++. Our theoretical results show that AA++ decreases the objective function faster than a uniform initialization in expectation. Furthermore, we (ii) argue that the $k$-means++ initialization can be seen as an approximation to our proposed strategy and that a Monte Carlo approximation to the $k$-means++ initialization can be adapted for AA++ for a more efficient initialization. Most importantly, we (iii) empirically demonstrate that our proposed initialization for archetypal analysis almost always outperforms all baselines on 15 real-world data sets. Lastly, (iv) our extensive empirical evaluation serves as the first comprehensive evaluation of different initialization strategies for AA which is of independent interest.

## 2 Preliminaries

Before introducing archetypal analysis, we briefly revisit $k$-means clustering since we will build upon similar ideas. We use the following compact notation: $[n] = \{1, 2, \ldots, n\}$ for an $n \in \mathbb{N}$.

**$k$-means Clustering.** Let $\mathcal{X} = \{\mathbf{x}_i\}_{i=1}^n \subset \mathbb{R}^d$ be a data set of $n$ points in $d$ dimensions, $\mathcal{Z} = \{\mathbf{z}_1, \ldots, \mathbf{z}_k\}$ be a set of $k$ cluster centers, and $d(\mathbf{x}, \mathcal{Z})^2 = \min_{\mathbf{q} \in \mathcal{Z}} \|\mathbf{x} - \mathbf{q}\|_2^2$ be the minimal squared Euclidean distance from a data point $\mathbf{x}$ to the closest center in $\mathcal{Z}$. The $k$-means clustering problem has the following objective:

$$\phi_{\mathcal{X}}(\mathcal{Z}) = \sum_{\mathbf{x} \in \mathcal{X}} d(\mathbf{x}, \mathcal{Z})^2 = \sum_{\mathbf{x} \in \mathcal{X}} \min_{\mathbf{q} \in \mathcal{Z}} \|\mathbf{x} - \mathbf{q}\|_2^2.$$

Often, the cluster centers $\mathcal{Z}$ of $k$-means are initialized using the $k$-means++ initialization procedure (Arthur & Vassilvitskii, 2007; Ostrovsky et al., 2013), which is guaranteed to yield a solution that is $\mathcal{O}(\log k)$-competitive with the optimal clustering. The idea is as follows. The first center is chosen uniformly at random. Then, the remaining $k-1$ cluster centers are chosen according to a probability distribution where the probability of choosing a point $\mathbf{x}$ is proportional to the minimal squared distance to the already chosen cluster centers, *i.e.*, $p(\mathbf{x}) \propto d(\mathbf{x}, \mathcal{Z})^2$. The procedure is outlined in Algorithm 3, which can be found in Appendix B.

**Archetypal Analysis.** Let $\mathcal{X} = \{\mathbf{x}_i\}_{i=1}^n \subset \mathbb{R}^d$ be a data set consisting of $n \in \mathbb{N}$ $d$-dimensional data points arranged as rows in the design matrix $\mathbf{X} \in \mathbb{R}^{n \times d}$. The idea of archetypal analysis (AA) (Cutler & Breiman, 1994) is to (approximately) represent every data point $\mathbf{x}_i$ as a convex combination of $k \in \mathbb{N}$ archetypes

$\mathcal{Z} = \{\mathbf{z}_1, \ldots, \mathbf{z}_k\}$, *i.e.*,

$$\mathbf{x}_i^\mathsf{T} \approx \mathbf{a}_i^\mathsf{T} \mathbf{Z}, \quad \mathbf{a}_i^\mathsf{T} \mathbf{1} = 1, \quad \mathbf{a}_i \geq 0, \tag{1}$$

where the matrix $\mathbf{Z} \in \mathbb{R}^{k \times d}$ contains the archetypes as rows and the vector $\mathbf{a}_i \in \mathbb{R}^k$ defines the weights for the $i$th data point. Here, $\mathbf{1}$ denotes the vector of ones and $\mathbf{a}_i \geq 0$ is meant element-wise. The archetypes $\mathbf{z}_j$ ($j \in [k]$) themselves are also represented (exactly) as convex combinations, but of the data points, *i.e.*,

$$\mathbf{z}_j^\mathsf{T} = \mathbf{b}_j^\mathsf{T} \mathbf{X}, \quad \mathbf{b}_j^\mathsf{T} \mathbf{1} = 1, \quad \mathbf{b}_j \geq 0,$$

where $\mathbf{b}_j \in \mathbb{R}^n$ is the weight vector of the $j$th archetype. Let $\mathbf{A} \in \mathbb{R}^{n \times k}$ and $\mathbf{B} \in \mathbb{R}^{k \times n}$ be the matrices consisting of the weights $\mathbf{a}_i$ ($i \in [n]$) and $\mathbf{b}_j$ ($j \in [k]$). Then, archetypal analysis yields an approximate factorization of the design matrix $\mathbf{X}$ as follows

$$\mathbf{X} \approx \mathbf{ABX} = \mathbf{AZ}, \tag{2}$$

where $\mathbf{Z} = \mathbf{BX} \in \mathbb{R}^{k \times d}$ is the matrix of archetypes. Due to the convexity constraints, the weight matrices $\mathbf{A}$ and $\mathbf{B}$ are row-stochastic. The weight matrices $\mathbf{A}$ and $\mathbf{B}$ are typically determined by minimizing the approximation error in Frobenius norm, resulting in the following optimization problem

$$\begin{aligned} \underset{\mathbf{A},\mathbf{B}}{\text{minimize}} \quad & \|\mathbf{X} - \mathbf{ABX}\|_\mathrm{F}^2 \\ \text{subject to} \quad & \mathbf{A1} = 1, \ \mathbf{A} \geq 0 \quad \text{and} \quad \mathbf{B1} = 1, \ \mathbf{B} \geq 0. \end{aligned} \tag{3}$$

This can be equivalently expressed as minimizing the sum of projections of the data points on the archetype-induced convex hull as follows

$$\|\mathbf{X} - \mathbf{AZ}\|_\mathrm{F}^2 = \sum_{\mathbf{x} \in \mathcal{X}} \min_{\mathbf{q} \in \mathrm{conv}(\mathcal{Z})} \|\mathbf{x} - \mathbf{q}\|_2^2, \tag{4}$$

where $\mathrm{conv}(\mathcal{Z})$ refers to the convex hull of the set $\mathcal{Z}$. An example of archetypal analysis and the projection errors is depicted in Figure 1. The optimization problem is a generalized low-rank problem (Udell et al., 2016), which is biconvex but not convex. Because it is biconvex, a local optimum can be found via an alternating optimization scheme as introduced by Cutler & Breiman (1994), such as the standard one outlined in Algorithm 4, which can be found in Appendix B. However, the quality of such a local optimum is directly dependent on the initialization.

**Archetype Initializations.** Popular ways of initializing the archetypes $\mathcal{Z}$ is by using data points chosen uniformly at random (called *Uniform*) or the FurthestSum procedure (Mørup & Hansen, 2010; 2012), but we also introduce FurthestFirst. Originally proposed for the metric $k$-center problem, the *FurthestFirst* algorithm (Gonzalez, 1985; Hochbaum & Shmoys, 1985) selects the first center/archetype uniformly at random and iteratively adds the point that is furthest away from the closest already selected center/archetype. Formally, the index of the next point is $j^{\mathrm{next}} = \arg\max_{i \in [n]}(\min_{\mathbf{q} \in \mathcal{Z}} \|\mathbf{x}_i - \mathbf{q}\|_2)$. Specifically for archetypal analysis, Mørup & Hansen (2010; 2012) propose a modification of FurthestFirst called *FurthestSum*, which sums over the distances of the already selected points, *i.e.*, $j^{\mathrm{next}} = \arg\max_{i \in [n]}\left(\sum_{\mathbf{q} \in \mathcal{Z}} \|\mathbf{x}_i - \mathbf{q}\|_2\right)$. To increase the performance, the first point, which was chosen uniformly at random, is usually discarded in the end and replaced by a new point chosen via the criteria outlined above.

Although not designed as an initialization strategy, we also mention *AAcoreset* (Mair & Brefeld, 2019) as it was used for the initialization of archetypes by Black et al. (2022) and Chapman et al. (2022). A coreset is a small subset of the original data set that allows for a more efficient training of archetypal analysis. The selection probability of every data point is given by $p(\mathbf{x}) \propto \|\mathbf{x} - \mu\|_2^2$, where $\mu$ is the mean of the data set $\mathcal{X}$.

## 3   Archetypal Analysis++

The idea of our proposed archetypal analysis++ (AA++) initialization procedure is very similar to the one of $k$-means++. We begin by choosing the first archetype $\mathbf{z}_1$ uniformly at random. The second archetype

---

**Algorithm 1** Archetypal Analysis++ Initialization

---

1: **Input:** Set of $n$ data points $\mathcal{X}$, number of archetypes $k$
2: **Output:** Initial set of $k$ archetypes $\mathcal{Z}$
3: Sample an index $i$ uniformly at random from $[n]$, *i.e.*, using $p(i) = n^{-1}$
4: Append $\mathbf{x}_i$ to $\mathcal{Z}$
5: **while** $|\mathcal{Z}| < k$ **do**
6:     Sample $i$ using $p(i) \propto \min_{\mathbf{q} \in \text{conv}(\mathcal{Z})} \|\mathbf{x}_i - \mathbf{q}\|_2^2$
7:     Append $\mathbf{x}_i$ to $\mathcal{Z}$
8: **end while**

---

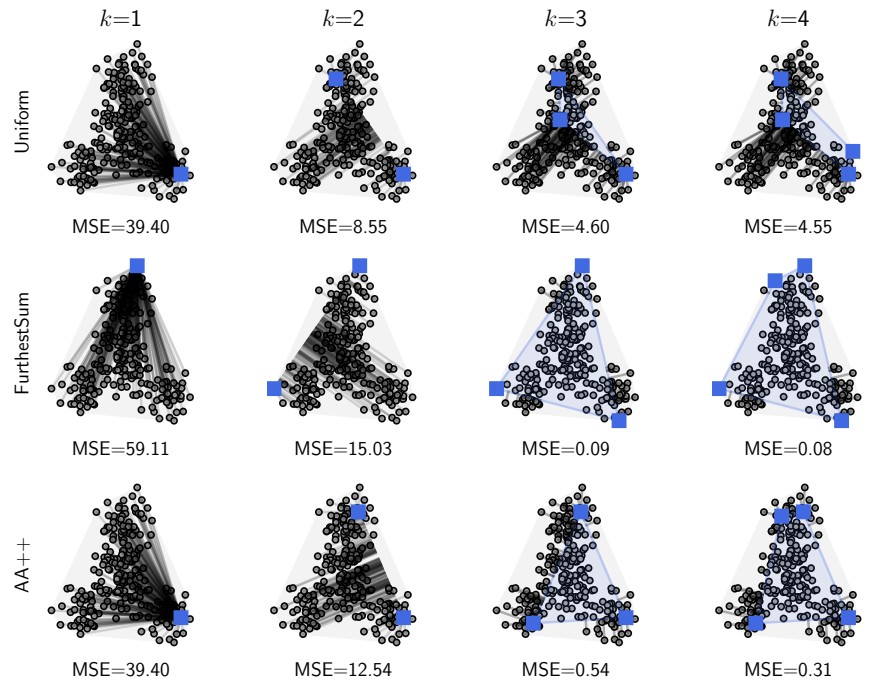

Figure 2: A comparison of Uniform, FurthestSum, and the proposed AA++ when consecutively initializing $k = 4$ archetypes. MSE denotes the mean square error, *i.e.*, Equation (4) multiplied by $n^{-1}$.

is chosen according to a distribution that assigns probabilities proportional to the distances from the first archetype, *i.e.*, $p(\mathbf{x}) \propto \|\mathbf{x} - \mathbf{z}_1\|_2^2$. The remaining $k - 2$ archetypes are chosen according to a probability distribution where the probability of choosing a point $\mathbf{x}$ is proportional to the minimum distance to the convex hull of the already chosen archetypes, *i.e.*, $p(\mathbf{x}) \propto \min_{\mathbf{q} \in \text{conv}(\{\mathbf{z}_1, \ldots, \mathbf{z}_k\})} \|\mathbf{x} - \mathbf{q}\|_2^2$. This procedure is outlined in Algorithm 1.

With every additional point sampled by AA++, the convex hull of the initialized factors, *i.e.*, $\text{conv}(\mathcal{Z})$, expands. This is because selecting a point outside the convex hull of $\mathcal{Z}$, *i.e.*, a point in $\{\mathbf{x} \in \mathcal{X} \mid \mathbf{x} \notin \text{conv}(\mathcal{Z})\}$, is by definition not contained in the convex hull of $\mathcal{Z}$ and hence expands it. In contrast, a point within the convex hull of $\mathcal{Z}$ would not increase its volume but has zero probability of being selected by AA++. Note that a uniform initialization also assigns probability mass to points in the interior of $\text{conv}(\mathcal{Z})$. Selecting a new point can make a previously selected point redundant, *i.e.*, it then lies in the convex hull of previously selected archetypes and does not help to increase its convex hull. However, this is not a problem in practice since redundant archetypes can be recovered during optimization.

Note that the optimization problem in line 6 of Algorithm 1 is the same as in Equation (1). Thus, parts of the archetypal analysis implementations can be re-used, simplifying the implementation of AA++. Besides, AA++ has no hyperparameters, and line 6 is trivially parallelizable.

Figure 2 depicts how Uniform, FurthestSum, and our AA++ strategy initialize $\mathcal{Z}$ on a synthetic data set. The gray projection errors can be seen as selection probabilities for AA++. In contrast, the selection probabilities of Uniform (not depicted) are not data-dependent, and FurthestSum sums the distances to the already selected archetypes up, *i.e.*, not the depicted distances. The sum of all depicted projections resembles Equation (4), and we provide the Mean Squared Error (MSE), *i.e.*, Equation (4) multiplied by $n^{-1}$, per iteration $k$.

In terms of MSE, FurthestSum works slightly better than AA++ on this synthetic data set. However, this is no longer the case with higher-dimensional real-world data, as our empirical evaluation later shows.

**Theoretical Analysis.** We first show that by adding a new archetype $\mathbf{z}$ to the set of archetypes $\mathcal{Z}$ in the while loop of AA++ in Algorithm 1, the objective function is guaranteed to decrease.

**Lemma 3.1.** *Let $\|\mathbf{X} - \mathbf{AZ}\|_{\mathrm{F}}^2 > 0$, i.e., there are points yielding projection errors. Then, adding a point $\mathbf{x} \in \mathcal{X} \setminus \mathcal{Z}$ to the set of archetypes $\mathcal{Z}$ according to AA++ (Algorithm 1) is guaranteed to decrease the objective function.*

Note that a uniform initialization does not have this guarantee since it also assigns probability mass to points that do not yield a reduction. We now show that, in expectation, AA++ reduces the objective function more than the uniform initialization; equality is just met if all candidate points have an equal projection distance.

**Proposition 3.2.** *Let $|\mathcal{Z}| \geq 1$ and $\|\mathbf{X} - \mathbf{AZ}\|_{\mathrm{F}}^2 > 0$. Per iteration, AA++ (Algorithm 1) yields a reduction of at least as much as the reduction for the uniform initialization in expectation. The reduction is equal only if every point in $\mathcal{X} \setminus \mathcal{Z}$ has the same projection onto the current set of archetypes $\mathcal{Z}$. Furthermore, using a uniform initialization cannot reduce the objective function more than AA++ in expectation.*

The proofs of both statements can be found in Appendix A.

**Complexity Analysis.** The proposed initialization strategy outlined in Algorithm 1 selects the first archetype uniformly at random. The remaining $k - 1$ archetypes are chosen according to a probability proportional to the squared distance between the candidate point and the convex hull of the already chosen archetypes. To compute this projection, a quadratic program (QP) has to be solved. Thus, the complexity of Algorithm 1 is $\mathcal{O}(n \cdot (k - 1) \cdot \mathrm{QP})$. It depends not only on the size of the data set $n$ and the number of archetypes $k$ to be initialized but also on the complexity of solving the quadratic program $\mathcal{O}(\mathrm{QP})$, which is often cubic in the number of variables $k$ (Goldfarb & Liu, 1990).

## 4   Approximating the Archetypal Analysis++ Initialization

The proposed initialization procedure AA++ has to solve $n \cdot (k - 1)$ quadratic programs, which can be time-consuming. Thus, we propose two strategies to approximate the initialization procedure. The first one approximates the distance computation (circumventing the computation of quadratic programs), and the second one approximates the sampling procedure (reducing $n$).

**Approximating the Distance Computation.** Mair & Brefeld (2019) show that the objective function of $k$-means upper bounds the objective of archetypal analysis, which is due to the per-point projections, *i.e.*,

$$\min_{\mathbf{q} \in \mathrm{conv}(\mathcal{Z})} \|\mathbf{x} - \mathbf{q}\|_2^2 \ \leq \ \min_{\mathbf{q} \in \mathcal{Z}} \|\mathbf{x} - \mathbf{q}\|_2^2$$

for a set of clusters/archetypes $\mathcal{Z}$. Since the computationally most expensive operation in the proposed Algorithm 1 is the projection onto the convex hull in line 6, it can be approximated by the distance to the closest point within the already chosen archetypes. See Figure 3 for an example. Note that the distance is then always over-estimated, and even points within the convex hull of the already chosen points might have a distance, although the projection should have a length of zero. This is depicted in Figure 3 for the red point. Following this approach boils down to the $k$-means++ initialization procedure. Hence, the new complexity is $\mathcal{O}(n \cdot k \cdot d)$, thus avoiding the cost of solving the QP.

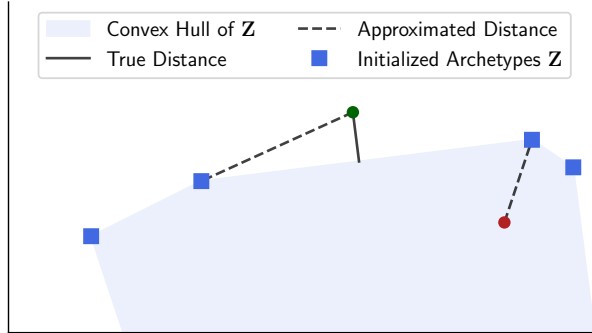

Figure 3: Approximation of the distance function in two dimensions. The true distance of the green point is depicted using a solid line whereas the approximation is shown as a (larger) dashed line. The red point has no distance to the convex hull, but the approximation yields a positive distance.

**Approximating the Sampling Procedure.** Another approach is to approximate the sampling procedure by not considering all $n$ data points, which is especially beneficial in large-scale scenarios. To initialize $k$-means++ in sublinear time, Bachem et al. (2016) leverage a Markov Chain Monte Carlo (MCMC) sampling procedure. This procedure is based on the Metropolis-Hastings algorithm (Hastings, 1970) with an independent and uniform proposal distribution. We adapt this idea for AA++ as follows. The first archetype is still sampled uniformly at random. For every following archetype, a Markov chain of length $m \ll n$ is constructed iteratively. We begin by sampling an initial point $\mathbf{x}_i$. Then, in every step of the chain, we sample a candidate point $\mathbf{x}_j$ and compute the acceptance probability, which is given by

$$\pi = \min\left(1, \frac{\min_{\mathbf{q} \in \text{conv}(\mathcal{Z})} \|\mathbf{x}_j - \mathbf{q}\|_2^2}{\min_{\mathbf{q} \in \text{conv}(\mathcal{Z})} \|\mathbf{x}_i - \mathbf{q}\|_2^2}\right).$$

With probability $\pi$, we update the current state from $\mathbf{x}_i$ to $\mathbf{x}_j$, otherwise we keep $\mathbf{x}_i$. After $m$ steps, we add the current $\mathbf{x}_i$ as the next archetype to the set of initial archetypes $\mathcal{Z}$. This approach is summarized in Algorithm 2, and the complexity of this strategy is $\mathcal{O}(m \cdot k \cdot \text{QP})$.

Bachem et al. (2016) also provide a theoretical result that bounds the error in terms of the total variation distance of the approximate sampling distribution to the true sampling distribution.

---

**Algorithm 2** AA++ Monte Carlo Initialization

1: **Input:** Set of $n$ data points $\mathcal{X}$, number of archetypes $k$, chain length $m$
2: **Output:** Initial set of $k$ archetypes $\mathcal{Z}$
3: Sample an index $i$ uniformly at random from $[n]$, *i.e.*, using $p(i) = n^{-1}$
4: Append $\mathbf{x}_i$ to $\mathcal{Z}$
5: **while** $|\mathcal{Z}| < k$ **do**
6:     Sample $i$ uniformly at random from $[n]$, *i.e.*, using $p(i) = n^{-1}$
7:     Compute the distance to the convex hull, *i.e.*, $d_i^2 = \min_{\mathbf{q} \in \text{conv}(\mathcal{Z})} \|\mathbf{x}_i - \mathbf{q}\|_2^2$
8:     **for** $l = 2, 3, \ldots, m$ **do**
9:         Sample $j$ uniformly at random from $[n]$, *i.e.*, using $p(j) = n^{-1}$
10:         Compute the distance to the convex hull, *i.e.*, $d_j^2 = \min_{\mathbf{q} \in \text{conv}(\mathcal{Z})} \|\mathbf{x}_j - \mathbf{q}\|_2^2$
11:         Sample $r$ uniformly at random from $(0, 1)$
12:         **if** $d_i^2 = 0$ or $d_j^2/d_i^2 > r$ **then**
13:             Set $i = j$ and $d_i^2 = d_j^2$
14:         **end if**
15:     **end for**
16:     Append $\mathbf{x}_i$ to $\mathcal{Z}$
17: **end while**

Here, $\|p - q\|_{\mathrm{TV}}$ denotes the total variation distance between two distributions $p$ and $q$ which is defined as

$$\|p - q\|_{\mathrm{TV}} = \frac{1}{2} \sum_{\mathbf{x} \in \Omega} |p(\mathbf{x}) - q(\mathbf{x})|,$$

where $\Omega$ is a finite sample space on which both distributions are defined on. The following bound on the error shows that the longer the chain length $m$, the smaller the error $\epsilon$.

**Theorem 4.1** (Adapted from Bachem et al. (2016)). *Let $k > 0$ and $0 < \epsilon < 1$. Let $p_{++}$ be the probability distribution over $\mathcal{Z}$ defined by using AA++ (Algorithm 1) and $p_{\mathrm{MCMC}}$ be the probability distribution over $\mathcal{Z}$ defined by using AA++MC (Algorithm 2). Then,*

$$\|p_{\mathrm{MCMC}} - p_{++}\|_{\mathrm{TV}} \leq \epsilon$$

*for a chain length $m = \mathcal{O}(\gamma' \log \frac{k}{\epsilon})$, where*

$$\gamma' = \max_{\mathcal{Z} \subset \mathcal{X}, |\mathcal{Z}| \leq k} \ \max_{\mathbf{x} \in \mathcal{X}} \ n \frac{d(\mathbf{x}, \mathcal{Z})^2}{\sum_{\mathbf{x}' \in \mathcal{X}} d(\mathbf{x}', \mathcal{Z})^2},$$

*and $d(\mathbf{x}, \mathcal{Z})^2 = \min_{\mathbf{q} \in \mathrm{conv}(\mathcal{Z})} \|\mathbf{x} - \mathbf{q}\|_2^2$.*

Note that $\gamma'$ is a property of the data set.

## 5   Experiments

**Data.** We use the following seven real-world data sets of varying sizes and dimensionalities. Additional eight real-world data sets, often smaller in size and dimension, are considered in Appendix D.

The *California Housing* (Pace & Barry, 1997) data set has $n = 20,640$ examples in $d = 8$ dimensions. A first large data set is *Covertype* (Blackard & Dean, 1999), which consists of $n = 581,012$ instances in $d = 54$ dimensions. *FMA*[1] (Defferrard et al., 2017) is a data set for music analysis that considers $n = 106,574$ songs represented with $d = 518$ features. It thus serves as a data set of higher dimensionality within our evaluation. *KDD-Protein*[2] has $n = 145,751$ data points, each represented with $d = 74$ dimensions measuring the match between a protein and a native sequence. The data set *Pose* is a subset of the Human3.6M data set (Catalin Ionescu, 2011; Ionescu et al., 2014). Pose was used in the ECCV 2018 PoseTrack Challenge and deals with 3D human pose estimation.[3] Each of the $n = 35,832$ poses is represented as 3D coordinates of 16 joints. Thus, the problem is 48-dimensional. The data set *RNA* (Uzilov et al., 2006) contains $n = 488,565$ RNA input sequence pairs with $d = 8$ features. Another larger data set we use is a subset of the Million Song Dataset (Bertin-Mahieux et al., 2011), which is called *Song*. It has $n = 515,345$ data points in $d = 90$ dimensions.

**Data Pre-processing.** We apply pre-processing to avoid numerical problems during learning and consider two different approaches: (i) *CenterAndMaxScale*, which centers the data and then divides the data matrix by its largest element, and (ii) *Standardization* which centers the data and then divides every dimension by its standard deviation. Results on *CenterAndMaxScale* are presented in the main body of the paper, while results on *Standardization* are presented in Appendix G for completeness.

**Baselines and Approximations.** We compare our proposed initialization strategy *AA++* against a *Uniform* subsample of all data points, *FurthestFirst* (Gonzalez, 1985; Hochbaum & Shmoys, 1985), *FurthestSum* (Mørup & Hansen, 2010; 2012), and *AAcoreset* (Mair & Brefeld, 2019). In addition, we evaluate our two proposed approximations of AA++. The first approximation, which is equivalent to *k-means++*, should thus not be seen as a baseline. Note that by $k$-means++ we only refer to the initialization strategy, *i.e.*, we do not run $k$-means afterwards. The second approximation uses a Markov Chain and is called *AA++MC*. We evaluate 1%, 5%, 10%, and 20% of the data set sizes as chain lengths $m$.

---

[1] https://github.com/mdeff/fma
[2] http://osmot.cs.cornell.edu/kddcup/datasets.html
[3] http://vision.imar.ro/human3.6m/challenge_open.php

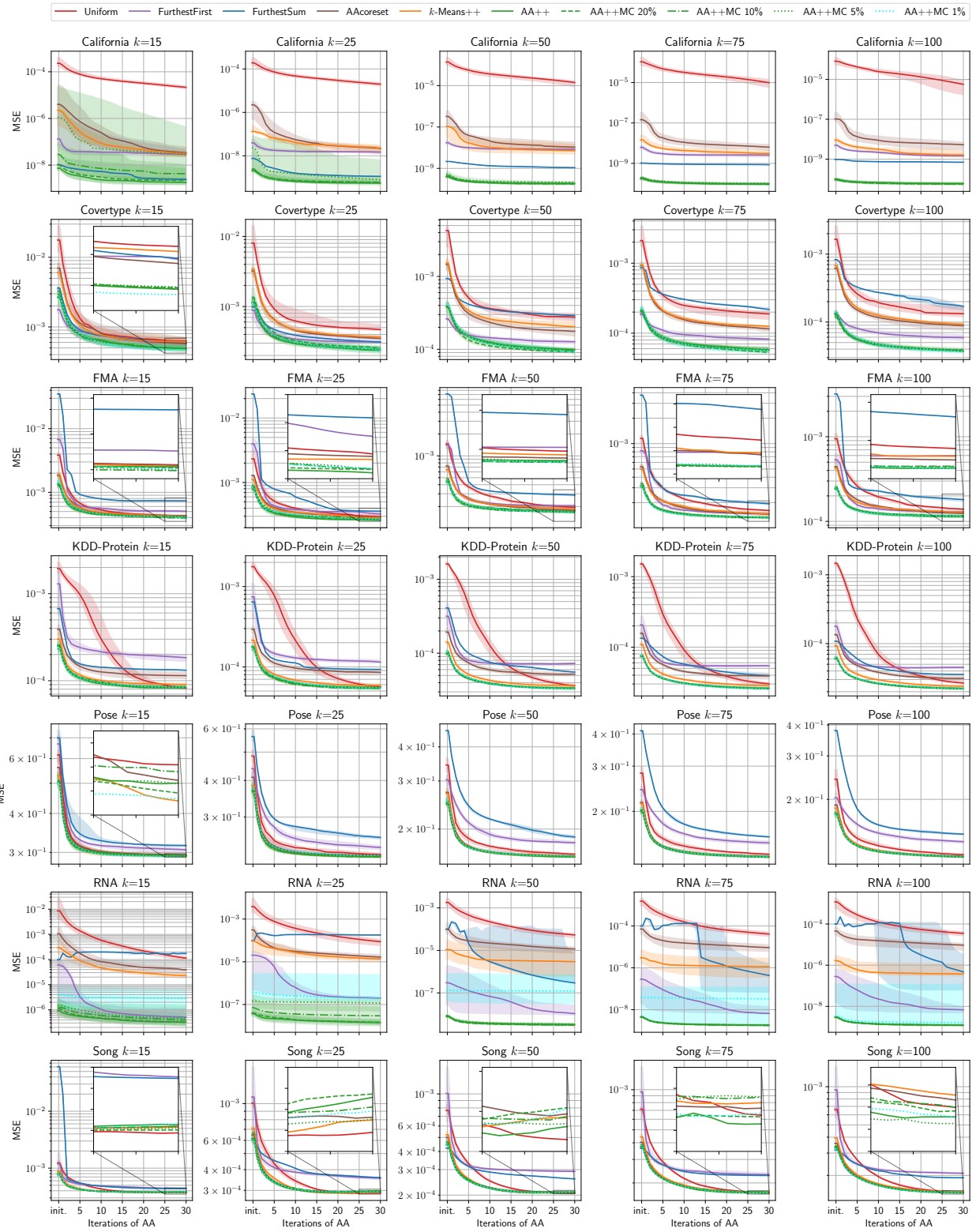

Figure 4: Results on California Housing, Covertype, FMA, KDD-Protein, Pose, RNA, and Song.

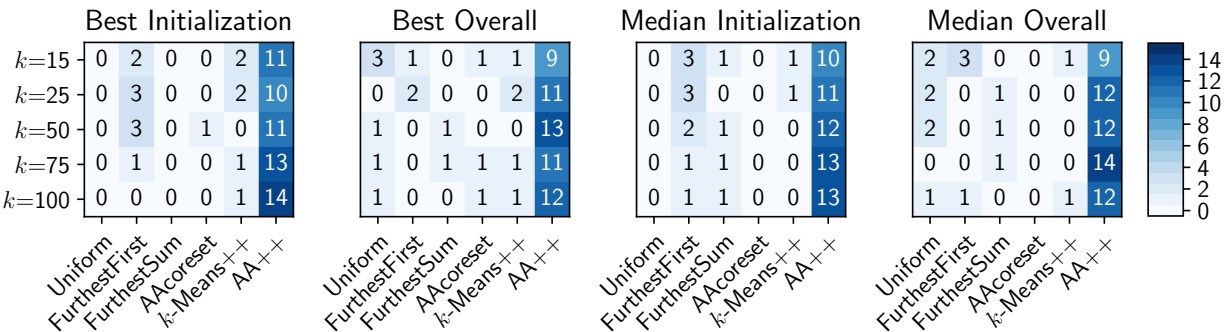

Figure 5: Aggregated statistics over 15 data sets (seven data sets from above and eight data sets from the appendix). Each table shows how often each initialization method yields the best result for various choices of $k$ under different settings. Best refers to the lowest error of a single seed and median refers to the median over many seeds. We report on the performance after initialization and overall during the optimization.

**Setup.** For various numbers of archetypes $k \in \{15, 25, 50, 75, 100\}$, we initialize archetypal analysis according to each of the baseline strategies and compute the Mean Squared Error (MSE), *i.e.*, the objective in Equation (4) normalized by $n^{-1}$. In addition, we perform a fixed number of 30 iterations of archetypal analysis based on those initializations. Here, we use the vanilla version of archetypal analysis according to Cutler & Breiman (1994). We compute statistics over 30 seeds, except for larger data sets ($n > 500,000$ or $d > 500$) for which we only compute 15 seeds and we report on median performances and depict the 75% and 25% quantiles. Furthermore, we track the time it takes to initialize the archetypes as well as the time each iteration of archetypal analysis takes. Details on the implementation are provided in Appendix C. All experiments run on an Intel Xeon machine with 28 cores with 2.60 GHz and 256 GB of memory.

**Performance Results.** Figure 4 depicts the results on California Housing, Covertype, FMA, KDD-Protein, Pose, RNA, and Song. Although being the most commonly used initialization methods, Uniform (red line) and FurthestSum (blue line) often yield the worst initializations, which can be seen by a short straight line on the left-hand side. As expected, the error decreases during the first 30 iterations of archetypal analysis. However, there is frequently a significant performance gap between Uniform when compared to AA++ (green line), especially for California Housing, Covertype, FMA, and RNA. The same gap is visible for FurthestSum on all data sets. The distance approximation - which is equivalent to $k$-means++ (orange line) - sometimes performs similarly to AA++ but is almost always better than Uniform. As for the Monte Carlo approximations of AA++ using 1%, 5%, 10%, and 20% of the data set size as chain lengths $m$, we can see that they approximate AA++ sufficiently well on these data sets, except on California Housing and RNA, where the smallest chain length struggles. Overall, our proposed AA++ initialization performs best in almost all cases.

This is confirmed by the aggregated statistics depicted in Figure 5. Each table shows how often each initialization method (MCMC approximations excluded) performs best on all evaluated data sets. In total, we consider 15 data sets: all previously introduced data sets and eight data sets, which are discussed in the appendix. Thus, 15 is the highest and best number that can appear in each of those tables. Within Figure 5, *best* refers to the lowest error among all seeds, and *median* refers to the lowest median over all seeds per method. Besides, *initialization* only considers the performance after initialization, whereas *overall* reflects the best performance across all 30 iterations of AA, which is typically at the last iteration. Once again, AA++ clearly yields the best results in all scenarios, especially for larger values of $k$. Note that we show $k$-means++ separately for completeness, but it should be seen as an approximation of AA++. For completeness, we also report on aggregated statistics on worst performances in Figure 8, which can be found in the appendix.

**Timing Results.** The increase in performance comes at a cost. Figure 6 depicts the time needed to initialize the archetypes on the California Housing and Covertype data sets for various choices of $k$. As expected, the fastest-performing method is Uniform since it uses no information about the data except the

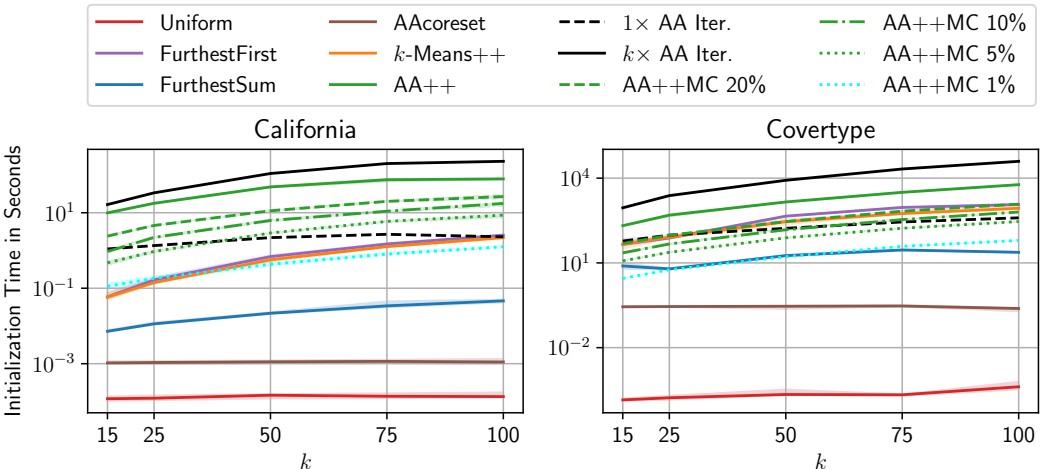

Figure 6: The median time it takes to initialize archetypal analysis.

number of data points. AAcoreset is slightly slower as it needs to compute the mean of the data as well as all distances to the mean. FurthestSum is slower than Uniform and AAcoreset, and the proposed approach AA++ is consistently the slowest. However, note that initializing $k$ points is still much cheaper than running $k$ AA iterations (black line). Using the distance approximation - which is equivalent to $k$-means++ - is much faster than AA++ and takes approximately as much time to initialize as FurthestFirst. Using the MCMC-based approximation also drastically reduces the initialization time of AA++. On Covertype, AA++MC using 1% of the data points as the chain length $m$ takes approximately as much time as FurthestSum, and other chain lengths $m$ lower the initialization time to the time of $k$-means++.

**Influence of the Pre-processing Scheme.** A surprising finding is that FurthestSum is especially sensitive to the pre-processing scheme. Standardization clearly often degrades the performance of FurthestSum but does not affect AA++ and its approximations as we show in Appendix G.

## 6   Discussion & Limitations

The idea of archetypal analysis++ is closely related to $k$-means++. Thus, it is natural to expect that a similarly strong theoretical guarantee, *i.e.*, $\mathbb{E}[\phi_{k\text{-means++}}] \leq \mathcal{O}(\log k)\phi_{\text{OPT}}$, can be derived for AA++ as for $k$-means++. Lloyds algorithm (Lloyd, 1982) consists of two rather simple steps that are iterated until convergence. First, within the assignment step, the distances between data points and cluster centers are computed, and each data point is assigned to its closest cluster center. Second, per cluster, the cluster centers are updated by computing the mean of all points assigned to this cluster. Note that those computations are relatively simple, and the second step can be performed in closed-form. Due to this simplicity, many theoretical analyses on $k$-means (including $k$-means++) exist. In contrast to that, archetypal analysis is a more challenging problem, not only computationally due to the optimization problems that do not admit a closed-form solution (*cf.* Algorithm 4), but also theoretically to analyze. For $k$-means, Arthur & Vassilvitskii (2007) exploit having closed-form solutions for the optimal cluster centers, and that only points in a cluster influence the position of their cluster center. This is not possible for archetypal analysis since there are no closed-form solutions, and there is an interdependence between archetypes since we are constructing a convex hull. Thus, we believe a similar strategy to derive theoretical guarantees for AA++ is unfruitful. However, note that neither of the state-of-the-art initializations for archetypal analysis come with any theoretical guarantees.

Undoubtedly, data-independent initialization strategies like Uniform – that chooses $k$ points from the data set uniformly at random – are the fastest way to initialize a model. Any data-dependent strategy will be more costly in terms of initialization time. One drawback of the proposed AA++ initialization strategy is that it is computationally more involved than its baselines. More specifically, it has to solve $n$ quadratic program

per archetype to be initialized except the first one. This renders the initialization of $k$ archetypes typically to be more expensive – in terms of time – than a single iteration of archetypal analysis but faster than $k$ iterations, *cf.* Figure 6. As a remedy, we proposed two approximations of AA++, which both decrease the initialization times to those of $k$-means++ or a single iteration of archetypal analysis. Thus, allowing for a practical usage of our proposed initialization strategy.

We might expect that the total runtime (30 iterations of archetypal analysis) is mainly determined by the different initialization efforts, *i.e.*, archetypal analysis is being the slowest when initialized using AA++. However, this is not true. As seen in, *e.g.*, Figure 10, the curves start indeed at different times, and AA++ is usually the most computationally demeaning initialization approach. Nonetheless, the total time of AA++ is not always the slowest, see, *e.g.*, Covertype in Figure 10. Note that the figure depicts the performance after initialization, followed by 30 interpolated points that correspond to the performances after 30 iterations of archetypal analysis. Per iteration, lines 5-8 in Algorithm 4 are computed. Specifically, line 5 involves solving $n$ non-negative least-squares problems to obtain the mixture matrix $\mathbf{A}$; line 6 computes the archetypes $\mathbf{Z}$ given $\mathbf{A}$ and the data matrix $\mathbf{X}$ by solving a system of linear equations; line 7 involves solving $k$ non-negative least-squares problems to obtain the mixture matrix $\mathbf{B}$; and line 8 computes the archetypes $\mathbf{Z}$ given $\mathbf{B}$ and $\mathbf{X}$ by computing a matrix-matrix product. There are several factors that determine the runtime of each and every iteration. First, to solve the non-negative least-squares problems, we utilize NNLS (Lawson & Hanson, 1995), which is an active set method that, depending on the specific inputs, needs a different number of internal iterations, and thus, the runtime of every non-negative least-squares problem varies. Second, instead of solving a system of linear equations in line 6, we solve a least-squares problem via `np.linalg.lstsq` for cases in which $\mathbf{A}^\mathsf{T}\mathbf{A}$ is not full-rank. Both factors can influence the runtime of an iteration of archetypal analysis.

Due to the bi-convex loss and the alternating optimization scheme, we expect the loss to decrease in every iteration. However, on rare occasions, *e.g.*, RNA and Song in Figure 4, the loss slightly increases. We conjecture that those cases are related to numerical instabilities. For example, instead of solving a system of linear equations in line 6 of Algorithm 4, we solve a least-squares problem via `np.linalg.lstsq` for cases in which $\mathbf{A}^\mathsf{T}\mathbf{A}$ is not full-rank. This might have an influence on the positioning of the archetypes $\mathbf{Z}$.

## 7 Related Work

Other variants exist besides the classical archetypal analysis (Cutler & Breiman, 1994). Moving archetypes (Cutler & Stone, 1997) define AA for moving targets. There are adaptations of archetypal analysis for missing data (Epifanio et al., 2020) and interval data (D'Esposito et al., 2012), and probabilistic archetypal analysis (Seth & Eugster, 2016) rephrases the factorization problem in a probabilistic way. De Handschutter et al. (2019) relax the convexity constraints of AA. More recently, approaches based on deep learning have been considered Keller et al. (2019); van Dijk et al. (2019); Keller et al. (2021). For those, the initialization of archetypes is irrelevant, as considered in this paper. Furthermore, archetypoid analysis (Vinué et al., 2015) restricts the archetypes to be data points instead of convex combinations of data points. Hence, the idea is similar to $k$-medoids (Kaufman & Rousseeuw, 1990) and intends to add even more interpretability. Note that our proposed AA++ can also be used as an initialization for archetypoid analysis.

Suleman (2017) stresses that an improper initialization of archetypal analysis is problematic and that the FurthestSum method is prone to selecting redundant archetypes, especially when having many archetypes. Nascimento & Madaleno (2019) consider an anomalous pattern initialization algorithm for initializing archetypal analysis. However, their focus was on inferring the number of archetypes $k$ and then using archetypal analysis for fuzzy clustering.

Less common initialization strategies for archetypal analysis include running $k$-means clustering, as used by Han et al. (2022), and utilizing a coreset (Mair & Brefeld, 2019), as used by Black et al. (2022) and Chapman et al. (2022). Note that the coreset was proposed as a way to condense the data set into a smaller set for a more efficient training of archetypal analysis rather than as a way for initializing it. However, we still include it as a baseline for completeness in our experimental evaluation. As for $k$-means, we drew inspiration from the $k$-means++ strategy but do not consider initialization strategies for $k$-means as suitable baselines for archetypal analysis. We rather focus on initialization strategies that are frequently used for archetypal analysis.

For the related non-negative matrix factorization (NMF) (Lee & Seung, 1999) problem, several initialization techniques based on randomization, other low-rank decompositions, clusterings, heuristics, and even learned approaches (Sjölund & Bånkestad, 2022) are used. A summary of various NMF initialization methods is provided by Esposito (2021). Among the considered baselines in this paper, all are randomized. However, FurthestSum can be seen as a heuristic algorithm, and FurthestFirst and $k$-means++ can be classified as (initializations of) clustering algorithms. The proposed AA++ approach is randomized just as Uniform, with the difference that the selection probabilities in AA++ are data-dependent while Uniform is data-independent.

## 8 Conclusion

We introduced archetypal analysis++ (AA++) as a new initialization method for archetypal analysis that is inspired by $k$-means++. The proposed method does not have any hyperparameters and is straightforward to implement, as it allows the reuse of already existing subroutines of any implementation of archetypal analysis. Furthermore, we proposed two approximations of AA++ that are faster to compute. The first one approximates the distance computation and is equivalent to the $k$-means++ initialization method. The second one is an MCMC approximation of the sampling procedure of AA++, which reduces the number of quadratic problems that have to be computed. We showed that both approximations work reasonably well and lower the initialization times to competitive levels. Empirically, we verified that, for two pre-processing schemes, AA++ almost always outperforms all baselines, including the most frequently used ones, namely Uniform and FurthestSum, on 15 real-world data sets of varying sizes and dimensionalities.

**Acknowledgments**

This work was partially supported by the Wallenberg AI, Autonomous Systems and Software Program (WASP) funded by the Knut and Alice Wallenberg Foundation. The authors thank the anonymous reviewers, Samuel G. Fadel, Ahcène Boubekki, Paul Häusner, and Zheng Zhao for valuable comments on earlier drafts.

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

# A  Proofs

## A.1  Proof of Lemma 3.1

**Lemma 3.1.** *Let $\|\mathbf{X} - \mathbf{AZ}\|_{\mathrm{F}}^2 > 0$, i.e., there are points yielding projection errors. Then, adding a point $\mathbf{x} \in \mathcal{X} \setminus \mathcal{Z}$ to the set of archetypes $\mathcal{Z}$ according to AA++ (Algorithm 1) is guaranteed to decrease the objective function.*

*Proof.* Let $\mathcal{P} = \mathrm{conv}(\mathcal{Z})$ be the polytope corresponding to the convex hull of the archetypes. There are only three cases. In the first case, the added point $\mathbf{x}$ lies within $\mathcal{P}$. Then, $\mathcal{P}$ remains unchanged, and so do the projections of the points outside of $\mathcal{P}$. Hence, the value of the objective function remains unchanged. However, AA++ (Algorithm 1) assigns a probability of zero to such a point $\mathbf{x}$. Due to $\|\mathbf{X} - \mathbf{AZ}\|_{\mathrm{F}}^2 > 0$, there has to be an $\mathbf{x} \in \mathcal{X}$ that has a projection error and thus reduces the objective function when picked. In the second case, the added point $\mathbf{x}$ lies outside of $\mathcal{P}$, and $\mathbf{x}$ is the only point projected on its face of $\mathcal{P}$. Then, the objective function decreases exactly by the projection of $\mathbf{x}$ onto $\mathcal{P}$. In the third case, the added point $\mathbf{x}$ lies outside of $\mathcal{P}$ and there are other points that lie on the same face of $\mathcal{P}$ as $\mathbf{x}$. Then, adding $\mathbf{x}$ decreases the objective function by the projection of $\mathbf{x}$ itself and also the projection of the other points that lie on the same face. Overall, using AA++ (Algorithm 1) decreases the objective function with probability one. $\square$

## A.2  Proof of Proposition 3.2

**Proposition 3.2.** *Let $|\mathcal{Z}| \geq 1$ and $\|\mathbf{X} - \mathbf{AZ}\|_{\mathrm{F}}^2 > 0$. Per iteration, AA++ (Algorithm 1) yields a reduction of at least as much as the reduction for the uniform initialization in expectation. The reduction is equal only if every point in $\mathcal{X} \setminus \mathcal{Z}$ has the same projection onto the current set of archetypes $\mathcal{Z}$. Furthermore, using a uniform initialization cannot reduce the objective function more than AA++ in expectation.*

Within the proof we use Chebyshev's sum inequality which states that if $p_1 \geq p_2 \geq \cdots \geq p_n$ and $r_1 \geq r_2 \geq \cdots \geq r_n$, then $\frac{1}{n} \sum_{i=1}^{n} p_i r_i \geq \left(\frac{1}{n} \sum_{i=1}^{n} p_i\right)\left(\frac{1}{n} \sum_{i=1}^{n} r_i\right)$.

*Proof of Proposition 3.2. Idea:* The cost that every point $\mathbf{x}_i$ contributes to the objective function is $d(\mathbf{x}_i, \mathcal{Z}) = \min_{\mathbf{q} \in \mathrm{conv}(\mathcal{Z})} \|\mathbf{x}_i - \mathbf{q}\|_2^2$. Hence, by adding $\mathbf{x}_i$ to $\mathcal{Z}$, the new objective function is reduced by at least $d(\mathbf{x}_i, \mathcal{Z})$. If $\mathbf{x}_i$ is the only point on its face of $\mathcal{Z}$, the reduction only involves $\mathbf{x}_i$ and hence the reduction is exactly $d(\mathbf{x}_i, \mathcal{Z})$. If there are more points on the same face as $\mathbf{x}_i$, choosing $\mathbf{x}_i$ reduces not only $d(\mathbf{x}_i, \mathcal{Z})$ but also the projection of any other point on that face. Hence, $d(\mathbf{x}_i, \mathcal{Z})$ serves as a lower bound on the reduction of the objective function when choosing $\mathbf{x}_i$. We now show that the reduction for AA++ is larger in expectation than for the uniform initialization.

Define $C = \sum_{i=1}^{n} d(\mathbf{x}_i, \mathcal{Z})$ which is equal to the normalization term within AA++. Without loss of generality, we can reorder the points such that $d(\mathbf{x}_1, \mathcal{Z}) \geq d(\mathbf{x}_2, \mathcal{Z}) \geq \cdots \geq d(\mathbf{x}_n, \mathcal{Z})$. Let $r_i = d(\mathbf{x}_i, \mathcal{Z})$ be the cost (potential reduction) the point $\mathbf{x}_i$ contributes to the objective function and $p_i = r_i/C$ be the probability of choosing the point $\mathbf{x}_i$ according to AA++. Thus, $r_1 \geq r_2 \geq \cdots \geq r_n$ and $p_1 \geq p_2 \geq \cdots \geq p_n$. Furthermore, let $u_i = \frac{1}{n}$ be the probability of choosing the point according to the uniform initialization. Due to Chebyshev's sum inequality, we obtain

$$\frac{1}{n} \sum_{i=1}^{n} p_i r_i \geq \left(\frac{1}{n} \sum_{i=1}^{n} p_i\right)\left(\frac{1}{n} \sum_{i=1}^{n} r_i\right) \iff \sum_{i=1}^{n} p_i r_i \geq \sum_{i=1}^{n} u_i r_i \iff \mathbb{E}_p[d(\mathbf{x}_i, \mathcal{Z})] \geq \mathbb{E}_u[d(\mathbf{x}_i, \mathcal{Z})],$$

by using $\sum_{i=1}^{n} p_i = 1$ and by multiplying both sides by $n$. Hence, choosing $\mathbf{x}_i$ according to AA++ (using the distribution $p$) results in a reduction which is at least as large as for a uniform initialization (using the distribution $u$). Equality is met, when $r_1 = r_2 = \cdots = r_n$, hence $p_1 = p_2 = \cdots = p_n$, and ultimately $p_i = u_i$ for all $i \in [n]$. If any point has a larger distance than the others, we have $r_1 > r_2$ due to the ordering and hence $p_1 > p_2$ in which the equality no longer holds and the reduction of the objective function is strictly larger for AA++ than for a uniform initialization in expectation. Note that due to the construction of $r_i$ and $p_i$, we can never have the case of flipping the inequality. In other words, AA++ cannot reduce the objective function less than that of a uniform initialization in expectation. $\square$

## B    Algorithms

Within the main body of the paper, we omitted algorithmic pseudocodes whenever the procedure is not as important or clear from the text. However, we now provide those algorithmic descriptions for completeness. Algorithm 3 outlines the $k$-means++ initialization procedure, and Algorithm 4 shows the standard alternating optimization scheme of archetypal analysis according to Cutler & Breiman (1994).

---

**Algorithm 3** $k$-means++ Initialization

---

1: **Input:** Set of $n$ data points $\mathcal{X}$, number of clusters $k$
2: **Output:** Initial set of $k$ clusters centers $\mathcal{Z}$
3: Sample index $i$ uniformly at random from $[n]$, *i.e.*, using $p(i) = n^{-1}$
4: Append $\mathbf{x}_i$ to $\mathcal{Z}$
5: **while** $|\mathcal{C}| < k$ **do**
6:     Sample $i$ using $p(i) \propto \min_{\mathbf{z} \in \mathcal{Z}} \|\mathbf{x}_i - \mathbf{z}\|_2^2$
7:     Append $\mathbf{x}_i$ to $\mathcal{Z}$
8: **end while**

---

---

**Algorithm 4** Archetypal Analysis (Cutler & Breiman, 1994)

---

1: **Input:** data matrix $\mathbf{X} \in \mathbb{R}^{n \times d}$, number of archetypes $k$
2: **Output:** weight matrices $\mathbf{A} \in \mathbb{R}^{n \times k}$ and $\mathbf{B} \in \mathbb{R}^{k \times n}$ and archetypes $\mathbf{Z} \in \mathbb{R}^{k \times d}$
3: Initialization of the archetypes $\mathbf{Z} \in \mathbb{R}^{k \times d}$, *e.g.*, via AA++
4: **while** not converged **do**
5:     $\mathbf{a}_i = \underset{\mathbf{a}_i^\mathsf{T}\mathbf{1}=1,\ \mathbf{a}_i \geq 0}{\arg\min} \|\mathbf{Z}^\mathsf{T}\mathbf{a}_i - \mathbf{x}_i\|_2^2 \quad \forall i \in [n]$
6:     $\mathbf{Z} = \underset{\mathbf{Z}}{\text{solve}}\ \mathbf{A}^\mathsf{T}\mathbf{A}\mathbf{Z} = \mathbf{A}^\mathsf{T}\mathbf{X}$
7:     $\mathbf{b}_j = \underset{\mathbf{b}_j^\mathsf{T}\mathbf{1}=1,\ \mathbf{b}_j \geq 0}{\arg\min} \|\mathbf{X}^\mathsf{T}\mathbf{b}_j - \mathbf{z}_j\|_2^2 \quad \forall j \in [k]$
8:     $\mathbf{Z} = \mathbf{B}\mathbf{X}$
9: **end while**

---

## C    Implementation Details

The code is implemented in Python using numpy (Harris et al., 2020) and is publicly available at `https://github.com/smair/archetypalanalysis-initialization`.

For the optimization problem within AA++ and archetypal analysis, *i.e.*, line 6 in Algorithm 1 and lines 5 and 7 in Algorithm 4, respectively, any QP solver can be used. In our implementation, we utilize the non-negative least-squares (NNLS) method from Lawson & Hanson (1995) and enforce the summation constraint by adding another equation in the linear system. For example, consider line 5 in Algorithm 4. There, we add a row of ones to $\mathbf{Z}^\mathsf{T}$ and a one to $\mathbf{x}_i$, *i.e.*, $\mathbf{1}^\top\mathbf{a} = 1 \cdot a_1 + 1 \cdot a_2 + \ldots + 1 \cdot a_k = 1$ to ensure that the vector $\mathbf{a}$ sums up to one. Specifically, we scale the ones by $M = 1000$ to have $1000 \cdot a_1 + 1000 \cdot a_2 + \ldots + 1000 \cdot a_k = 1000$ such that the other lines of the linear system $\mathbf{Z}^\mathsf{T}\mathbf{a}_i = \mathbf{x}_i$ do not dominate the *summation to one* constraint. Note that using NNLS with a large constant $M$ was already suggested by Cutler & Breiman (1994). Specifically, we use the Fortran implementation of NNLS from scipy[4], *i.e.*, `scipy.optimize.nnls`, and slightly adapt it by increasing the hard coded maximum number of iterations since NNLS occasionally ran out of iterations.

## D    Results on Additional Data Sets

We further evaluate the initialization methods on the following eight data sets. *Airfoil* (Brooks et al., 1989) has $n = 1,503$ data points represented in $d = 5$ dimensions. *Concrete* (Yeh, 1998) has $n = 1,030$ instances in

---

[4]`https://scipy.org/`

Table 1: An overview of the 15 data sets used in this paper. The upper part is considered in the main body of the paper and the lower part is discussed in the appendix.

| | Data Set Name | Number of Data Points | Number of Dimensions |
|---|---|---|---|
| Main Paper | California Housing | 20,640 | 8 |
| | Covertype | 581,012 | 54 |
| | KDD-Protein | 145,751 | 74 |
| | Pose | 35,832 | 48 |
| | RNA | 488,565 | 8 |
| | Song | 515,345 | 90 |
| | FMA | 106,574 | 518 |
| Appendix | Airfoil | 1,503 | 5 |
| | Concrete | 1,030 | 8 |
| | Banking1 | 4,971 | 7 |
| | Banking2 | 12,456 | 8 |
| | Banking3 | 19,939 | 11 |
| | Ijcnn1 | 49,990 | 22 |
| | MiniBooNE | 130,064 | 50 |
| | SUN Attribute | 14,340 | 102 |

$d = 8$ dimensions. The data sets *Banking1*, *Banking2*, and *Banking3* (Dulá & López, 2012) have $n = 4,971$, $n = 12,456$, and $n = 19,939$ points in $d = 7$, $d = 8$, and $d = 11$ dimensions, respectively. The *Ijcnn1* data set has $n = 49,990$ points in $d = 22$ dimensions and was used in the IJCNN 2001 neural network competition.[5] We employ the same pre-processing as Chang & Lin (2001). *MiniBooNE* (Dua & Graff, 2017) consists of $n = 130,064$ data points in $d = 50$ dimensions. Finally, the *SUN Attribute* (Patterson & Hays, 2012) has $n = 14,340$ data points in $d = 102$ dimensions. A summary of all used data sets is provided in Table 1.

Figure 7 depicts the performance for the additional eight data sets. Note that we omit AA++MC 1% for small data sets, *i.e.*, if $n < 25,000$. Again, we can see that Uniform and the FurthestSum initializations often perform worse than the proposed approach and its approximation. AA++ is almost always consistently best, except on some rare occasions, such as for $k = 25$ on Ijcnn1. While the MCMC approximations of AA++ often perform very close to AA++ itself, we can see some more significant performance gaps on some of these additional data sets. Especially on BankProblem1, most Monte Carlo versions fail to approximate AA++ properly. However, note that all approximations are still better than the Uniform baseline.

Figure 8 summarizes aggregated statistics over 15 data sets, similar to Figure 5. The difference is that we now analyze the worst instead of the best-case setting. After initialization, the most frequently used initialization strategies for archetypal analysis, namely Uniform and FurthestSum, often yield the highest errors and, thus, the worst initializations. Overall, they sometimes recover from the sub-optimal initializations but still perform worse more often than the proposed AA++ strategy.

## E   Timing Results

The results depicted in Figures 4 and 7 show the MSE as a function of iterations. However, as shown in Figure 6, the initialization times vary across all initialization strategies, and we should not assume that the time of an AA iteration is static, as discussed in Section 6. Thus, we also report on the MSE as a function of time in Figures 10 and 11. Surprisingly, although AA++ takes longer to initialize, the overall optimization time for 30 AA iterations is smaller for AA++ than for Uniform on Covertype, Concrete, and MiniBooNE data. A similar scenario can be seen for small $k$ values on Song data. On some other data sets, using AA++ takes more overall, mainly due to the slower initialization. However, in turn, we get better performance. Furthermore, using an approximation of AA++ yields a reasonable compromise time-wise while still being close to AA++ performance-wise.

---

[5]https://www.csie.ntu.edu.tw/~cjlin/libsvmtools/datasets/

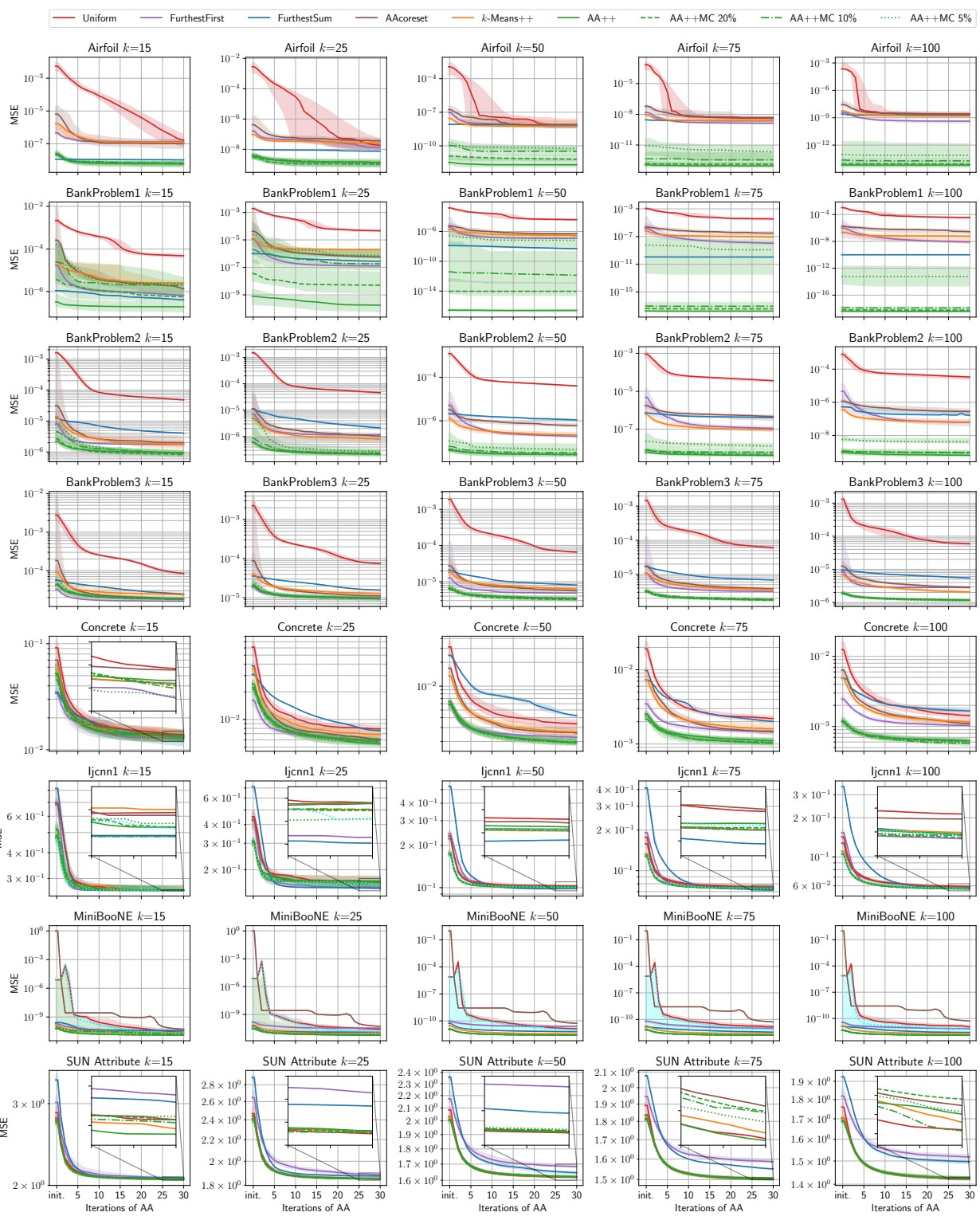

Figure 7: Results on Airfoil, Concrete, Banking1, Banking2, Banking3, Ijcnn1, MiniBooNE, and SUN Attribute using the *CenterAndMaxScale* pre-processing as in the main body of the paper.

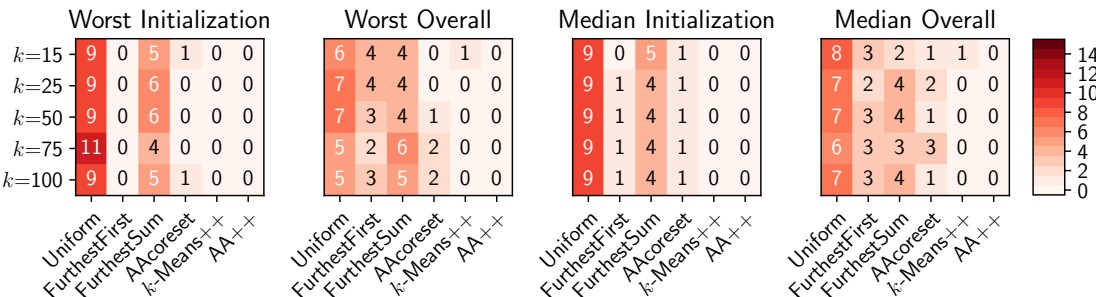

Figure 8: Aggregated statistics over 15 data sets (seven data sets from above and eight data sets from the appendix) using *CenterAndMaxScale* as pre-processing. Each table shows how often each initialization method yields the worst result for various choices of $k$ under different settings. Worst refers to the highest error of a single seed and median refers to the median over many seeds. We report on the performance after initialization and overall during the optimization.

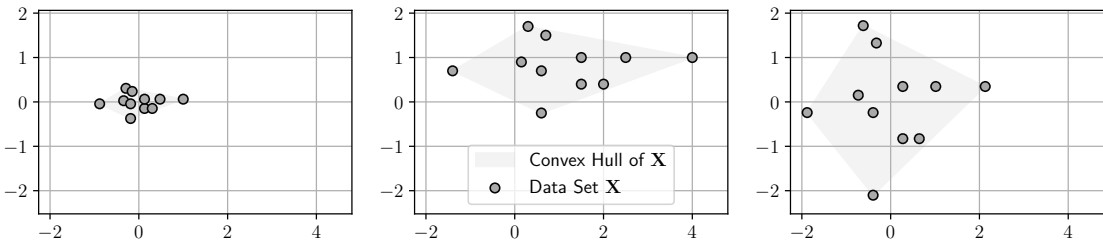

Figure 9: Comparison between the two pre-processing approaches. Left: Center data and divide by maximum value. Middle: Original data set. Right: Standardized data set.

## F    Pre-processing

In the main body of the paper, we pre-processed the data by first centering the data set and then dividing it by the maximum value. Another frequently used pre-processing scheme is standardization. That is, per dimension, we subtract the mean and divide it by the standard deviation. Since both are linear transformations, they do not change the membership of the points being on the border of the convex hull (Ziegler, 2012). However, the former scheme maintains the shape of the data set in terms of its convex hull, whereas the latter scheme changes it. This is depicted in Figure 9, where the original data is shown in the middle, the pre-processing of the main body of the paper is shown on the left-hand side and a data standardization is shown on the right-hand side.

## G    Results on Standardized Data Sets

We also conduct the same set of experiments on all data sets using standardization as pre-processing. In Figure 12, we can see for the first set of data sets that FurthestSum usually performs worst, often by a large margin. In contrast, the most consistent behavior has the proposed AA++ method, which is often the best. Besides, the proposed approximations of AA++ perform sufficiently close to AA++ itself. Figure 13 shows the same results but with respect to time instead of iterations. The performance of the second set of data sets is depicted in Figure 14 and the version showing time is provided in Figure 15.

Aggregated statistics on the results using standardization as pre-processing are depicted in Figure 16. As expected, AA++ wins on most of the data sets irrespective of the applied setting. However, the numbers are slightly lower than in Figure 5, which summarizes the results using the CenterAndMaxScale pre-processing.

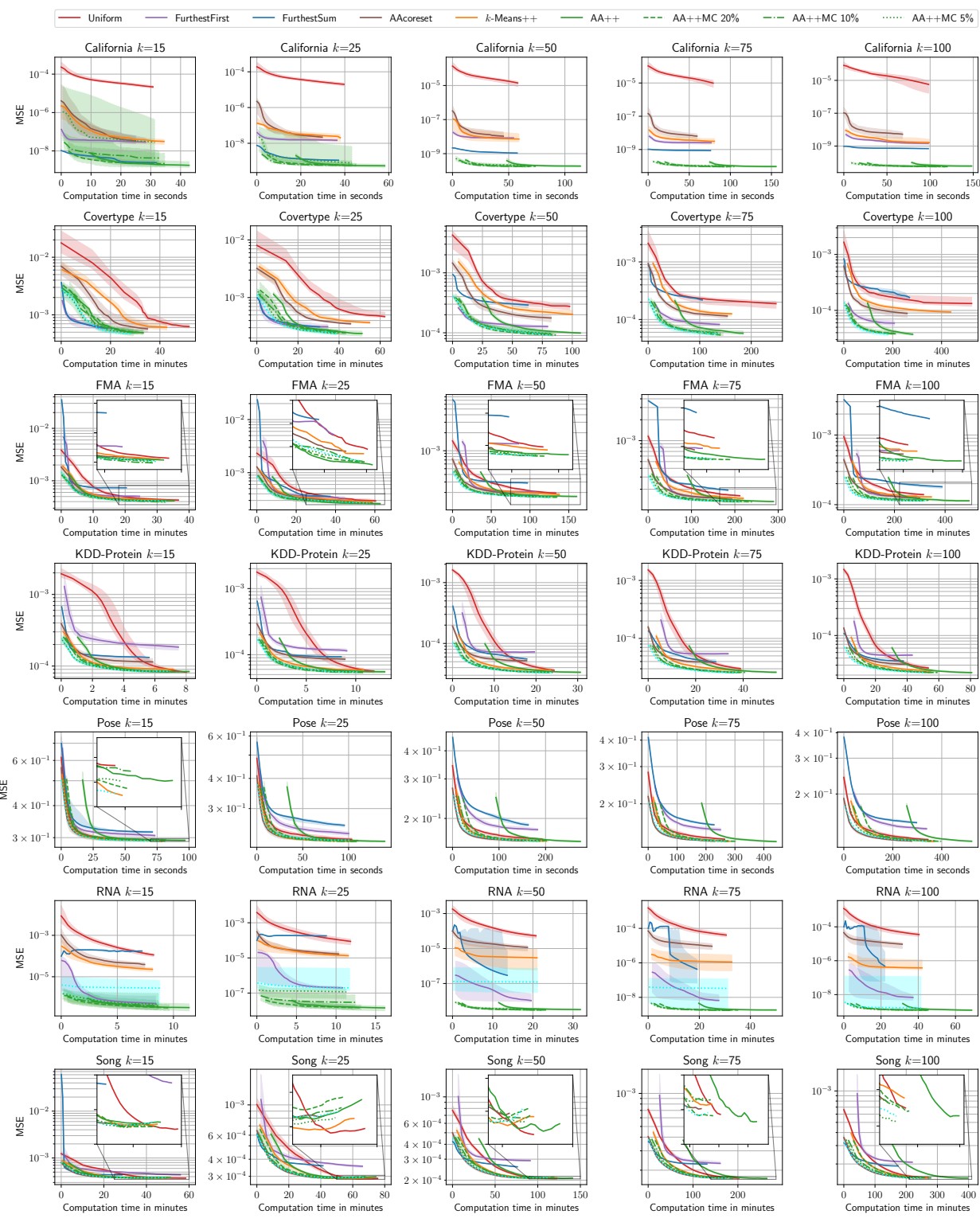

Figure 10: Results on California Housing, Covertype, FMA, KDD-Protein, Pose, RNA, and Song. In contrast to Figure 4, this figure shows the computation time of the initialization followed by 30 iterations of archetypal analysis on the $x$-axis.

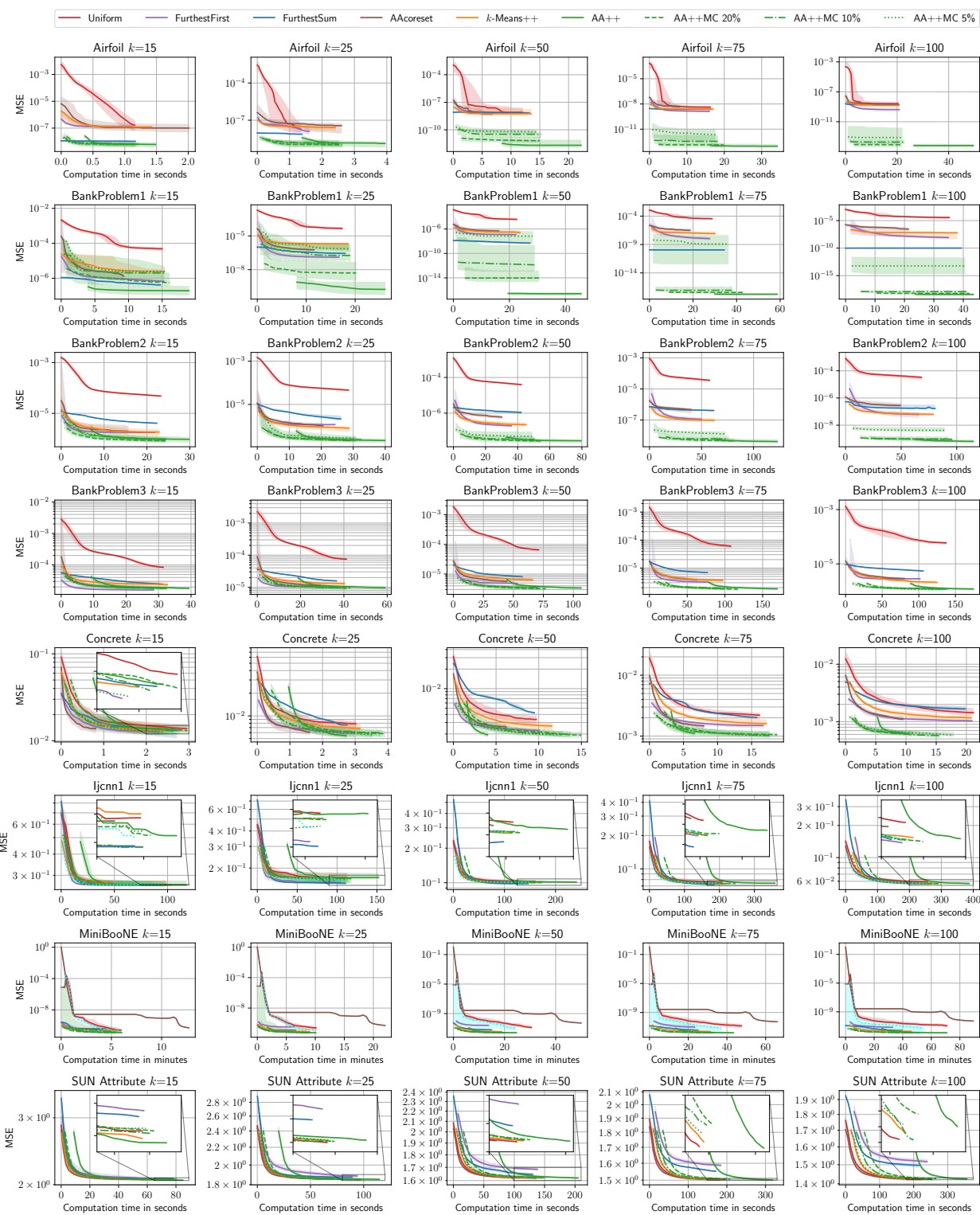

Figure 11: Results on Airfoil, Concrete, Banking1, Banking2, Banking3, Ijcnn1, MiniBooNE, and SUN Attribute using the *CenterAndMaxScale* pre-processing as in the main body of the paper. In contrast to Figure 7, this figure shows the computation time of the initialization followed by 30 iterations of archetypal analysis on the *x*-axis.

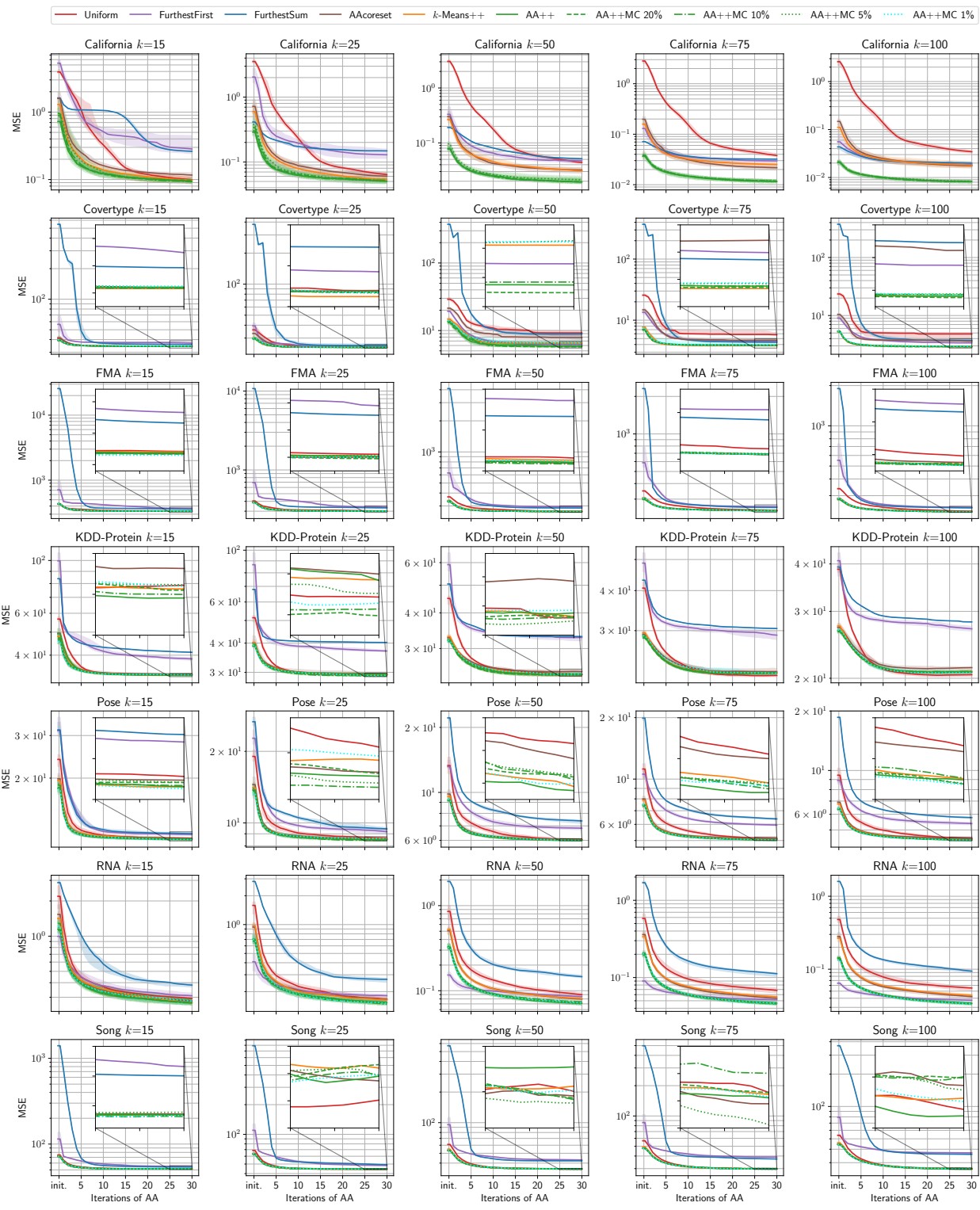

Figure 12: Results on California Housing, Covertype, FMA, KDD-Protein, Pose, RNA, and Song using *Standardization* as pre-processing.

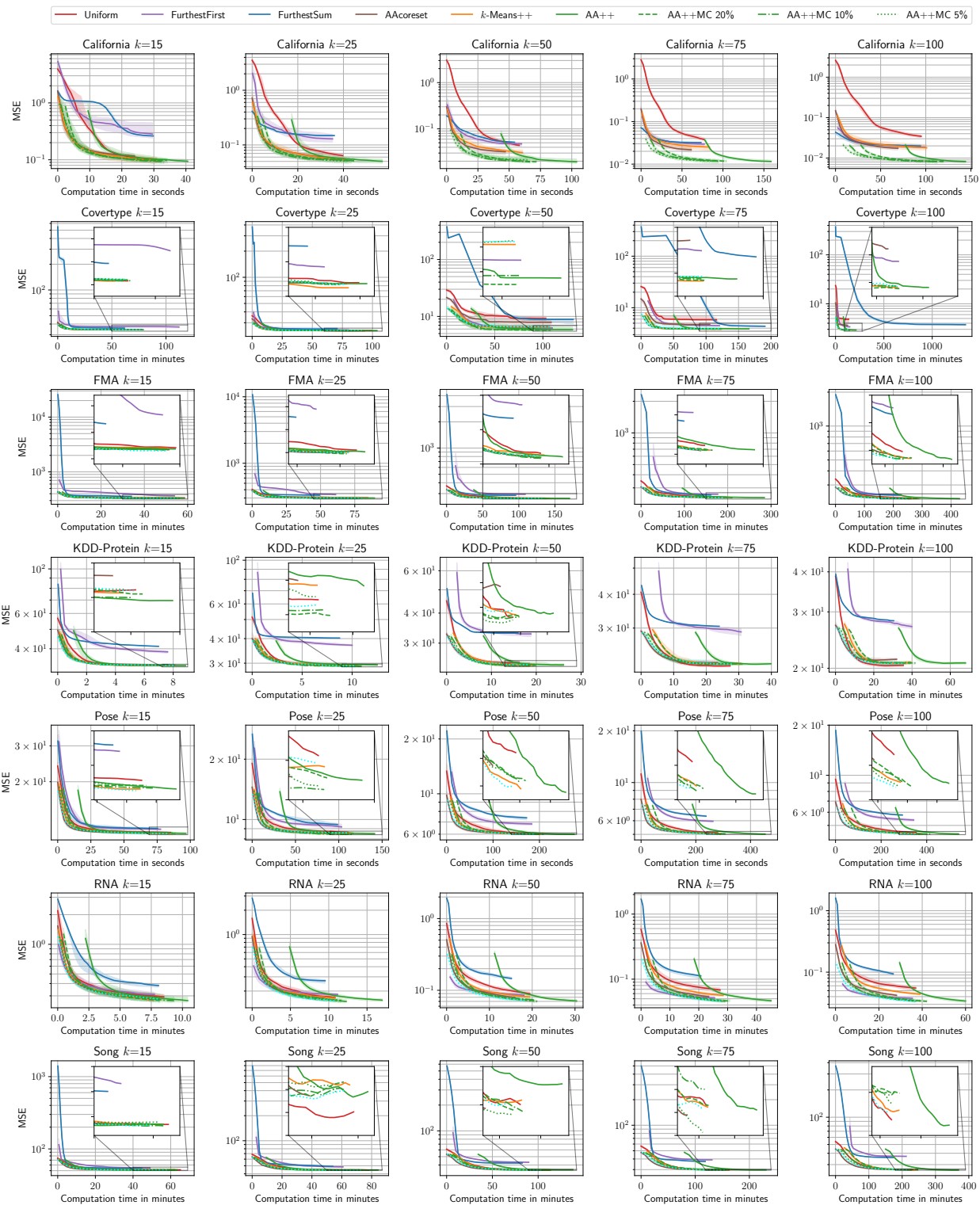

Figure 13: Results on California Housing, Covertype, FMA, KDD-Protein, Pose, RNA, and Song using *Standardization* as pre-processing. In contrast to Figure 12, this figure shows the computation time of the initialization followed by 30 iterations of archetypal analysis on the *x*-axis.

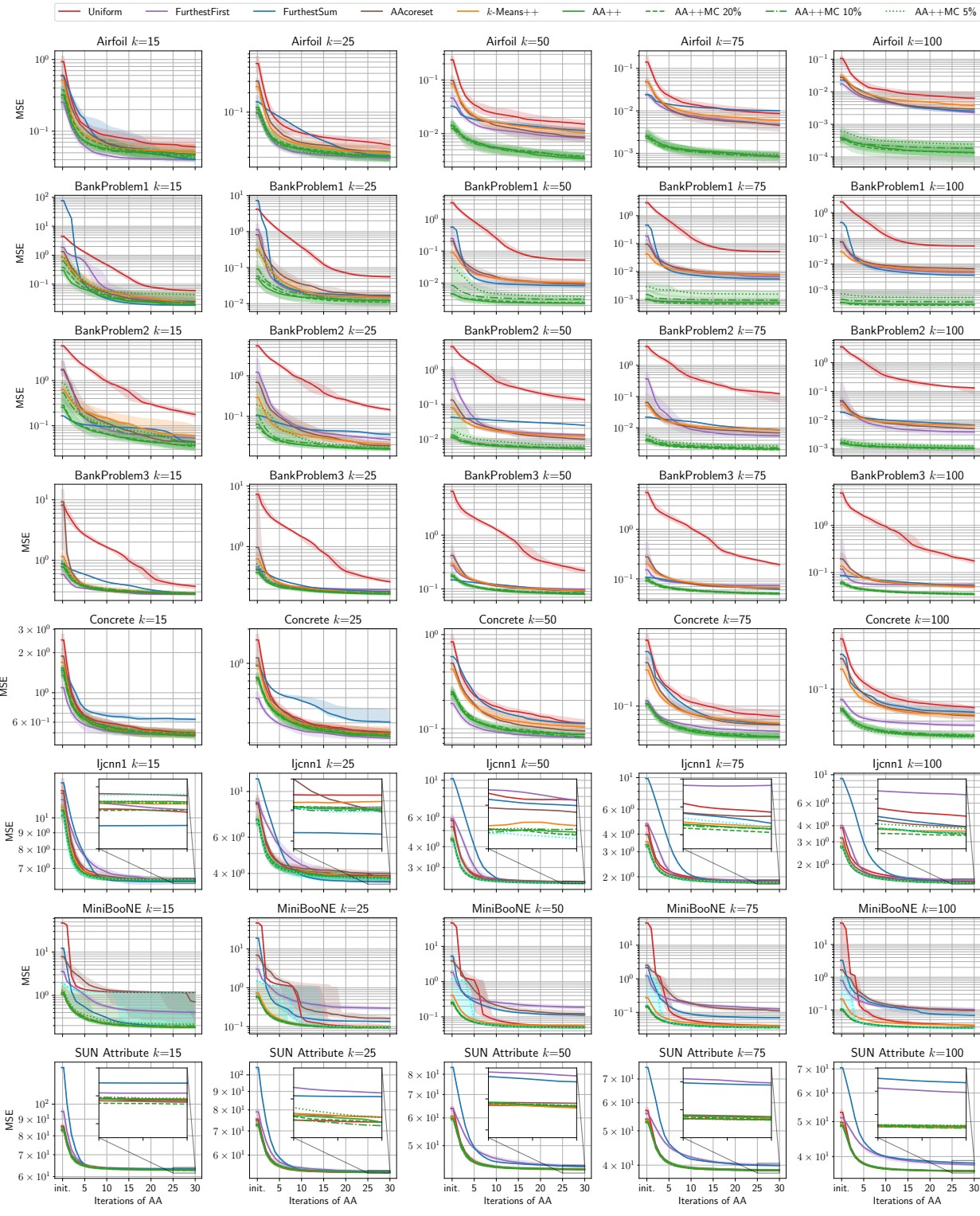

Figure 14: Results on Airfoil, Concrete, Banking1, Banking2, Banking3, Ijcnn1, MiniBooNE, and SUN Attribute using *Standardization* as pre-processing.

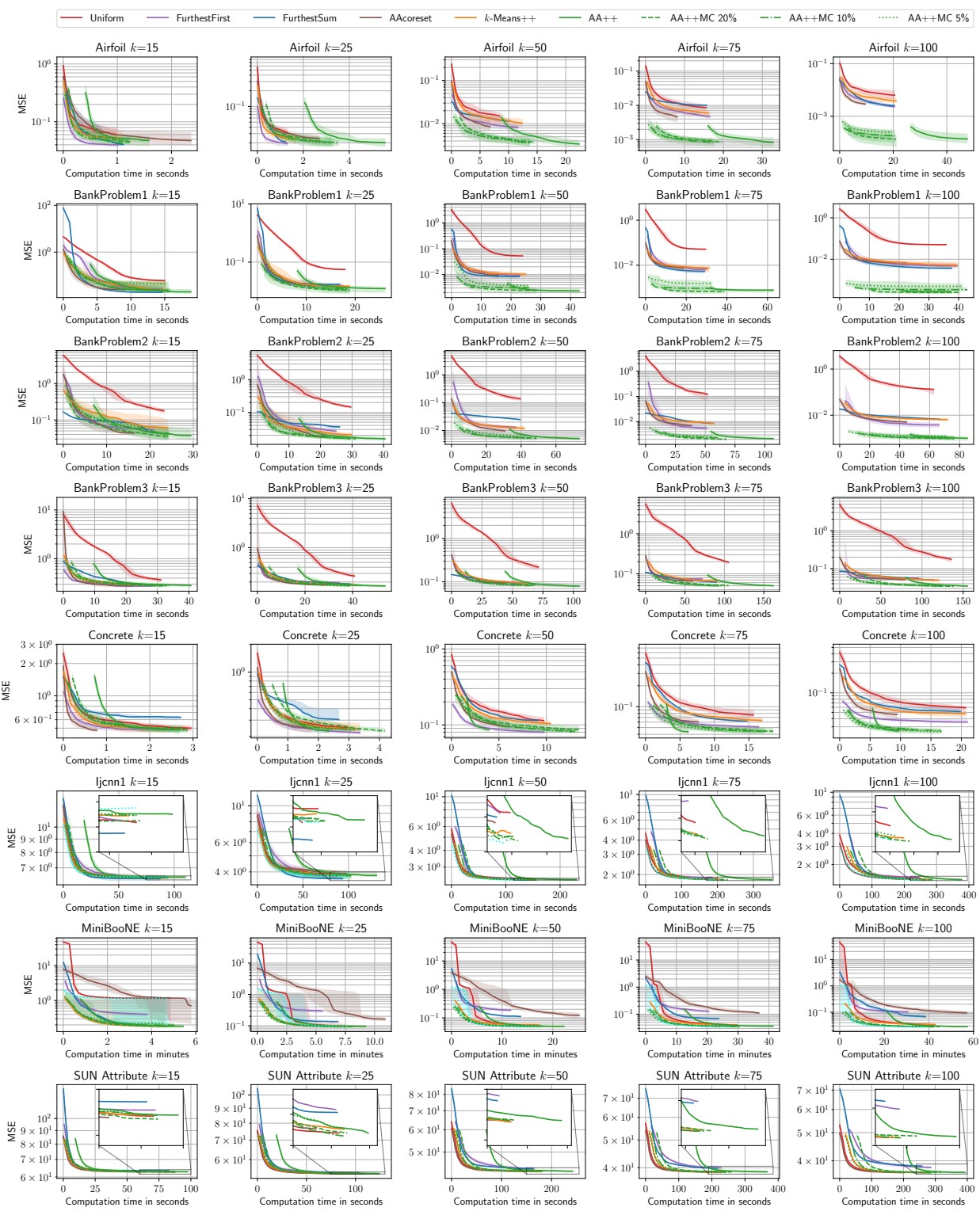

Figure 15: Results on Airfoil, Concrete, Banking1, Banking2, Banking3, Ijcnn1, MiniBooNE, and SUN Attribute using *Standardization* as pre-processing. In contrast to Figure 14, this figure shows the computation time of the initialization followed by 30 iterations of archetypal analysis on the $x$-axis.

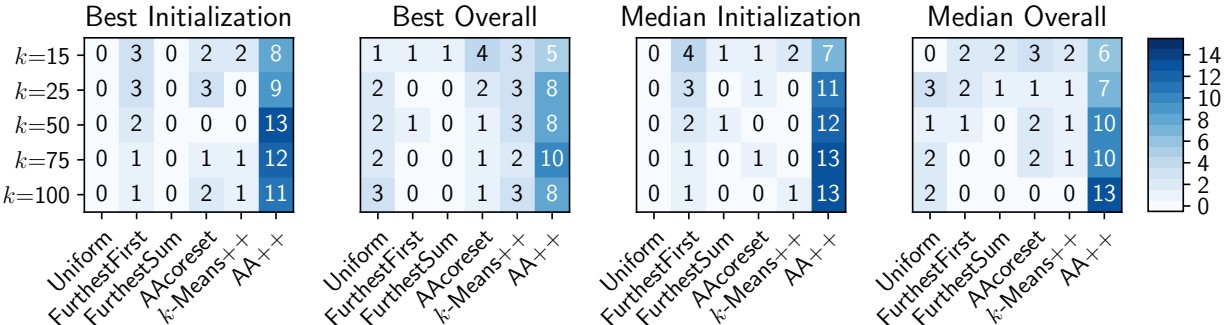

Figure 16: Aggregated statistics over 15 data sets (seven data sets from the main paper and eight data sets from the appendix) using *Standardization* as pre-processing. Each table shows how often each initialization method yields the best result for various choices of $k$ under different settings. Best refers to the lowest single seed and median refers to the median over many seeds. We report on the performance after initialization and overall during the optimization.

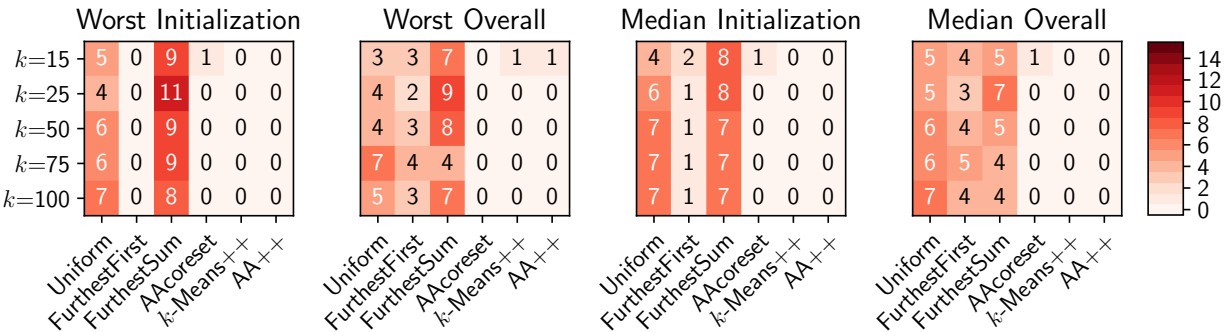

Figure 17: Aggregated statistics over 15 data sets (seven data sets from the main paper and eight data sets from the appendix) using *Standardization* as pre-processing. Each table shows how often each initialization method yields the worst result for various choices of $k$ under different settings. Worst refers to the highest error of a single seed and median refers to the median over many seeds. We report on the performance after initialization and overall during the optimization.

