# OpenReview forum: "Archetypal Analysis++: Rethinking the Initialization Strategy"
_TMLR — Accepted by TMLR_

### Review · Reviewer_n3Fp · 2024-02-27

**Summary Of Contributions:**

This paper proposes a novel initialization scheme for AA (Archetypal Analysis), called AA++.

Given a dataset (x_1, \dots, x_n), AA seeks $k$ anchor points $z_1, \dots z_k$ that are each a convex combination of the $x_i$, so that each $x_i$ can be well approximated by a convex combination of the $z_j$. The loss function to minimize is $\min_{A, B} \|X - ABX\|^2$ where $A$ and $B$ are entry-wise positive and row-wise stochastic.

This is a non-convex problem so initialization is of extreme importance. AA++ works in a similar fashion as k-means ++: the first point $z_1$ is taken uniformly among the dataset points, and then samples iteratively $z_{j}$ following a distribution proportionnal to $p(i)\propto  \min_{q\in\mathrm{Conv}(z_j)}\|x_i - q\|^2$.

Computing each $p(i)$ is a quadratic program, hence the authors propose a cheaper version of the algorithm, by sampling only from a subset of points at each iteration.

The authors then compare their approach to several baselines on 14 datasets, and demonstrate the promises of their approach.

**Audience:**

Yes

**Claims And Evidence:**

Yes

**Requested Changes:**

In my view, either improving the theory of the paper or explaining clearly what is the limiting factor in extending the kmeans++ bounds on the loss would enhance the paper.

**Strengths And Weaknesses:**

Strengths:
- The paper is extremely well written and is pleasant to read. I learned a lot from reading it. The figures are very helpful.
- The proposed method is natural and sound
- It indeed seems to work better than all the other considered baselines
- The theoretical results look correct
- The experiments are instructive and clearly show the benefits of the proposed methods.

Weaknesses

The main weakness of that paper, in my view, is its theory. I find the theoretical part underwhelming. The authors propose an initialization scheme quite similar in spirit to k-means ++, yet they do not discuss similar theoretical results as the ones obtained in the original kmeans++ paper. The reader is left expecting a mention on the expected value of the loss function after initialization, as done in kmeans++. If the extension of the proof is hard and beyond the scope of the paper, then I think it should be mentioned in the paper, as well as the precise difficulties encountered by the authors.

I have some other minor concerns:

- In the experiments, some curves go up, doesn't the AA algorithm have a descent guarantee?
- The algorithm used to solve the least-squares under simplex constraint could be explained in further detail, as it is a critical part of both AA and AA++ and is -- if I'm not mistaken -- the main bottleneck of the methods.

Misc:
- In fig2, recalling the MSE definition or at least pointing to it would help the reader
- I think that fig 9 gives a more interesting picture than fig 4, since it clearly shows both the superiority of the proposed method and the advantage of the MC methods.
- A parallel could be made between FurtherFirst and the proposed method in the following two ways. AA++ can be seen as a relaxation of furtherfirst, where instead of sampling from the argmax of a distribution the authors sample from the distribution itself. It can also be connected to a different loss function; indeed, in kmeans++, the authors mention that to optimize the loss $\|X - ABX\|^l_2$ then, one should sample from $ \|X - ABX\|^l_2$. If I'm not mistaken, FurtherFirst therefore corresponds to the $l\to \infty$ limit, being the optimal initialization for a cost function which is not the MSE.

---

> ### Author Response · Authors · 2024-03-25
>
> We thank the reviewer for the careful reading and the valuable comments.
>
> > The main weakness of that paper, in my view, is its theory. I find the theoretical part underwhelming. The authors propose an initialization scheme quite similar in spirit to k-means ++, yet they do not discuss similar theoretical results as the ones obtained in the original kmeans++ paper. The reader is left expecting a mention on the expected value of the loss function after initialization, as done in kmeans++. If the extension of the proof is hard and beyond the scope of the paper, then I think it should be mentioned in the paper, as well as the precise difficulties encountered by the authors.
> > In my view, either improving the theory of the paper or explaining clearly what is the limiting factor in extending the kmeans++ bounds on the loss would enhance the paper.
>
> We agree that it is unfortunate that AA++ lacks a theoretical guarantee such as the one of $k$-means++ where they bound the value of the objective function after initialization by a scaled objective function value of the optimal clustering.
> To explain why, let's first recall what $k$-means is and how it works. Lloyds algorithm consists of two steps:
>
> 1. Assignment step: Compute the distances between points and cluster centers and choosing per point the cluster center that is closest.
> 2. Update step: Per cluster, update the cluster center by computing the mean of all points assigned to this cluster.
>
> Note that those computations are rather simple and the second step can be performed in closed-form. Due to this simplicity, many theoretical analyses on $k$-means (including $k$-means++) have been published.
> In contrast to that, archetypal analysis is a harder problem, not only computationally due to the optimization problems that do not admit a closed-form solution (lines 5 and 7 in Algorithm 4), but also theoretically to analyze.
> Now, consider Lemma 3.1 in "$k$-means++: The Advantages of Careful Seeding" by Arthur & Vassilvitskii. The authors exploit (i) having closed-form solutions for the optimal cluster centers and that (ii) only points in a cluster influence the position of their cluster center. As mentioned, this is not possible for archetypal analysis since (i) there are no closed-form solutions and (ii) there is an interdependence between archetypes since we are constructing a convex hull.
>
> Furthermore, note that neither of the state-of-the-art initializations for archetypal analysis, Uniform and FurthestSum, come with any theoretical guarantees (on the objective function value similar to those of $k$-means++).
>
> We will add a paragraph after Proposition 3.2 explaining why a similar guarantee as for $k$-means++ is hard to obtain for AA++. The arguments will be similar as the explanations above.
>
> > In the experiments, some curves go up, doesn't the AA algorithm have a descent guarantee?
>
> We agree that this behavior is not expected. Since the loss is bi-convex, we should expect to descent in every iteration. Note that the cases in which the objective function increases are rare. We conjecture that those cases are related to numerical instabilities. For example, instead of solving a system of linear equations in line 6 of Algorithm 4, we solve a least-squares problem via `np.linalg.lstsq` for cases in which $A^\top A$ is not full-rank. This might have an influence on the positioning of the archetypes $Z$.

---

> ### Author Response · Authors · 2024-03-25
>
> > The algorithm used to solve the least-squares under simplex constraint could be explained in further detail, as it is a critical part of both AA and AA++ and is -- if I'm not mistaken -- the main bottleneck of the methods.
>
> The reviewer is correct that the computation of the non-negative least squares problems is the bottleneck, not only of our proposed AA++ (line 6 in Algorithm 1), but also for archetypal analysis (lines 5 and 7 in Algorithm 4) in general. Note that any optimization algorithm solving this problem can be used.
>
> Specifically, we utilize the NNLS method as stated on the bottom of page 7 and we use the implementation of scipy, i.e., `scipy.optimize.nnls`. Note that this used to be a wrapper that calls a function implemented in FORTRAN. Only recently (we just found out ourselves as we write this rebuttal), scipy switched to a python implementation of NNLS. However, we are still using the FORTRAN implementation of NNLS and slightly adapted it: We increased the hardcoded maximum number of iterations since occasionally NNLS ran out of iterations. To fulfill the summation constraint, we add a line of ones to the system of equations, i.e., $1^\top a = 1 \cdot a_1+1 \cdot a_2+\ldots+1 \cdot a_k=1$ to ensure that the vector $a$ sums up to one. Specifically, we scale the one-vector by $M=1000$ to have $1000 \cdot a_1+1000 \cdot a_2+\ldots+1000 \cdot a_k=1000$ such that the other lines of the linear system do not dominate the *summation to one* constraint. Note that using NNLS with this large constant $M$ was already suggested by Cutler and Breiman, (1994).
>
> We clarify the implementation specifics in the appendix.
>
> > In fig2, recalling the MSE definition or at least pointing to it would help the reader
>
> The meaning of MSE is explained in the text when discussing Figure 2 at the very beginning of page 5 as "The sum of all depicted projections resembles Equation (4) and we provide the Mean Squared Error (MSE), i.e., Equation (4) normalized by $n^{-1}$, per iteration." However, we can also add this detail in the caption of Figure 2.
>
> > I think that fig 9 gives a more interesting picture than fig 4, since it clearly shows both the superiority of the proposed method and the advantage of the MC methods.
>
> The reviewer has a good point and we are willing to swap Figures 4 and 9 if the action editor agrees.
>
> > A parallel could be made between FurtherFirst and the proposed method in the following two ways. AA++ can be seen as a relaxation of furtherfirst, where instead of sampling from the argmax of a distribution the authors sample from the distribution itself. It can also be connected to a different loss function; indeed, in kmeans++, the authors mention that to optimize the loss $\|X-ABX\|^l_2$ then, one should sample from $\|X-ABX\|^l_2$. If I'm not mistaken, FurtherFirst therefore corresponds to the $l \to \infty$ limit, being the optimal initialization for a cost function which is not the MSE.
>
> Using $\|X-ABX\|^l_2$ (and also taking the convexity constraints into consideration) as a loss is essentially using the Euclidean projection per point to the convex hull of the archetypes raised to the power of $l$. If $l \to \infty$, the point with the largest projection distance determines the loss. However, FurthestFirst does not take the convexity constraints into account. Thus, at some point during initialization, FurthestFirst also starts selecting points in the middle of the data set. This is beneficial for a clustering task, but not for initializing archetypal analysis, which rather cares about boundary points of a data set. Note that AA++ never selects a point for initialization that is within the convex hull of the already selected points.

---

> > ### Comment · Reviewer_n3Fp · 2024-03-25
> > **Thanks for your replay**
> >
> > I thank the authors for their reply. I have read the other reviewer's comments and the response by the authors. I agree with the points raised by the other reviewers, and I believe that the authors addressed all of our concerns. in my view this paper is clearly worthy of acceptance to TMLR.
> >
> > I find the discussion regarding the lack of better theoretical guarantees very interesting, I think it deserves a spot in the main text.
> > The numerical instabilities leading to the non-monotonicity of the curves should also be mentioned.

---

> > > ### Author Response · Authors · 2024-03-26
> > >
> > > Thank you for your response. We are glad that we could address your concerns and questions.

---

### Review · Reviewer_tPNN · 2024-03-11

**Summary Of Contributions:**

The paper studies the initialization strategy for archetypal analysis, which is a matrix factorization method with many applications in data science. The paper introduces the archetypal analysis++ initialization method. The basic idea is to build a distribution per iteration, which gives more weight to those examples far away from the convex hull of the existing archetype. Then, sample the archetypal from this distribution and append it. To further relieve the computational cost, the paper further proposes two methods: one avoids the computation of the distance to the convex hull, and the second uses MCMC to relax the dependency on the number of examples. The paper gives comprehensive experiments to show the behavior of the proposed initialization method.

**Audience:**

Yes

**Broader Impact Concerns:**

I have no broader impact concerns.

**Claims And Evidence:**

Yes

**Requested Changes:**

- It would be helpful if the authors can add discussions why they consider the specific archetypal analysis method in Algorithm 4. The authors mentioned that each iteration in Algorithm 4 is expensive, and may be more expensive than the cost of the initialization, which justifies the importance of initialization. Suppose we consider a more efficient archetypal analysis method with less computational cost per iteration. Then is the initialization still important in this case?
- Section 5: "The same gape" should be "The same gap"
- Section 6: "as a heuristic" should be "as a heuristic algorithm"
- Algorithm 4 requires to compute the inverse of $A^TA$. Is this matrix always invertible?

**Strengths And Weaknesses:**

Strength:
- The paper is very clearly written. Although I am not familiar with archetypal analysis, I can still follow the paper clearly.
- The paper includes very comprehensive experimental results. Experimental results include the comparison with existing baseline initialization methods as measured by MSE, the comparison on time and the aggregated statistics on the number of times the proposed method yields the best results. The paper considers the effect of the preprocessing strategy and the number of archetypal. The experimental results seem to be convincing.

Weakness:
- The theoretical results are not strong. The theoretical analysis only shows that the proposed method is no worse than the uniform initialization in expectation.
- The proposed initialization method is time consuming. Fortunately, this problem is relaxed as the paper introduces two approximate methods.
- The paper mainly considers the initialization method. While good initialization can improve the performance, a general belief is that the later archetypal analysis may be more important. In the experimental results, the paper only uses Algorithm 4 as the archetypal analysis method, which was proposed 30 years ago. Suppose we consider a more advanced archetypal analysis method, can we still get similar results showing the advantage of the proposed initialization method?

---

> ### Author Response · Authors · 2024-03-25
>
> We thank the reviewer for the careful reading and the valuable comments.
>
> > The theoretical results are not strong.
>
> We agree that it is unfortunate that AA++ lacks a theoretical guarantee such as the one of $k$-means++ where they bound the value of the objective function after initialization by a scaled objective function value of the optimal clustering.
> To explain why, let's first recall what $k$-means is and how it works. Lloyds algorithm consists of two steps:
>
> 1. Assignment step: Compute the distances between points and cluster centers and choosing per point the cluster center that is closest.
> 2. Update step: Per cluster, update the cluster center by computing the mean of all points assigned to this cluster.
>
> Note that those computations are rather simple and the second step can be performed in closed-form. Due to this simplicity, many theoretical analyses on $k$-means (including $k$-means++) have been published.
> In contrast to that, archetypal analysis is a harder problem, not only computationally due to the optimization problems that do not admit a closed-form solution (lines 5 and 7 in Algorithm 4), but also theoretically to analyze.
> Now, consider Lemma 3.1 in "$k$-means++: The Advantages of Careful Seeding" by Arthur & Vassilvitskii. The authors exploit (i) having closed-form solutions for the optimal cluster centers and that (ii) only points in a cluster influence the position of their cluster center. As mentioned, this is not possible for archetypal analysis since (i) there are no closed-form solutions and (ii) there is an interdependence between archetypes since we are constructing a convex hull.
>
> Furthermore, note that neither of the state-of-the-art initializations for archetypal analysis, Uniform and FurthestSum, come with any theoretical guarantees (on the objective function value similar to those of $k$-means++).
>
> We will add a paragraph after Proposition 3.2 explaining why a similar guarantee as for $k$-means++ is hard to obtain for AA++. The arguments will be similar as the explanations above.
>
> > The theoretical analysis only shows that the proposed method is no worse than the uniform initialization in expectation.
>
> This is actually not true. Our theoretical analysis (Proposition 3.2) shows that **our approach is better than uniform in expectation**. Equal performance can only be achieved in special (almost constructed) cases. While Uniform can sample a better initialization than AA++, this is not possible in expectation, because in expectation AA++ cannot perform worse than Uniform.
>
> Note that no other initialization strategy comes with any theoretical guarantee.
>
> > The paper mainly considers the initialization method.
>
> We exclusively consider the initialization of archetypal analysis.
>
> > While good initialization can improve the performance, a general belief is that the later archetypal analysis may be more important.
>
> Can the reviewer please provide a reference for this statement? Depending on the specific sub-optimal initialization, archetypal analysis is either trapped in a local minima from which it cannot recover (see, e.g., FurthestSum in Figure 4), while, for example, the sub-optimal initialization of Uniform for KDD-Protein in Figure 4 does recover. This is exactly the reason why we compute and show 30 iterations of archetypal analysis after initialization. However, as the reviewer can see in many figures, especially in Figures 5, 8, 16 and 17, our proposed approach overall yields the best results.

---

> ### Author Response · Authors · 2024-03-25
>
> > In the experimental results, the paper only uses Algorithm 4 as the archetypal analysis method, which was proposed 30 years ago.
>
> We are unsure why the publication date of archetypal analysis matters. Archetypal analysis is a learning problem, i.e., it can be formulated as an optimization problem consisting of an objective function to be minimized and a set of constraints. There are several ways on how to solve this optimization problem. We chose the standard way and stick to it for all experiments. Besides, we are unsure why this choice should matter.
>
> > Suppose we consider a more advanced archetypal analysis method, can we still get similar results showing the advantage of the proposed initialization method?
>
> Can the reviewer please specify what a "more advanced archetypal analysis method" is? We propose a better initialization for archetypal analysis as it is. If one changes the learning problem, the initialization should most probably be adjusted depending on the severity of the changes.
>
> > It would be helpful if the authors can add discussions why they consider the specific archetypal analysis method in Algorithm 4.
>
> We are not using a specific archetypal analysis but rather **the archetypal analysis**, i.e., the default/original way of optimizing it and not an approximation or other variant of it.
>
> > The authors mentioned that each iteration in Algorithm 4 is expensive, and may be more expensive than the cost of the initialization, which justifies the importance of initialization. Suppose we consider a more efficient archetypal analysis method with less computational cost per iteration. Then is the initialization still important in this case?
>
> Again, we are unsure what the reviewer means. What is a more efficient archetypal analysis? Why should a computationally more efficient implementation be less vulnerable to a sub-optimal initialization?
>
> > Section 5: "The same gape" should be "The same gap"
> >
> > Section 6: "as a heuristic" should be "as a heuristic algorithm"
>
> Thank you for pointing out those issues. We will address them.
>
> > Algorithm 4 requires to compute the inverse of $A^\top A$. Is this matrix always invertible?
>
> Not necessarily. Instead of solving a system of linear equations in line 6, we actually solve a least-squares problem via `np.linalg.lstsq` which covers the cases in which $A^\top A$ is not full-rank.

---

> > ### Comment · Reviewer_tPNN · 2024-03-26
> > **Thank you for your response**
> >
> > I would like the authors for their point-to-point response to my comments. My concerns are well addressed and have no other queries.

---

> > > ### Author Response · Authors · 2024-03-26
> > >
> > > Thank you for your response. We are glad that we could address your concerns and questions.

---

### Review · Reviewer_fwRK · 2024-03-18

**Summary Of Contributions:**

This paper proposes a novel initialization approach, called AA++, for archetypal analysis. The proposed method leverages the idea of kmeans++ and has better performance than uniform initialization. The authors also propose two strategies to approximate AA++ and significantly reduce the computational costs.

**Audience:**

Yes

**Broader Impact Concerns:**

No broader impact concerns.

**Claims And Evidence:**

Yes

**Requested Changes:**

1. For the theory, I would like to see more elaborate analysis for AA++ and AA++MC, as that of k-means++ (Arthur & Vassilvitskii, 2007) and k-MC^2 (Bachem et al. (2016).
2. For the experiments, I would like to see both time comparison and performance comparison on datasets with larger dimension (d>=500).

The first suggestion is critical to my recommendation for acceptance and the second suggestion can strengthen the work in my view.

**Strengths And Weaknesses:**

Strengths:

1. The paper theoretically shows that AA++ outperforms the uniform initialization.
2. The empirical results show that the proposed initialization can help AA to converge to a better minima in many cases.

Weaknesses:

1. The theoretical contribution is a bit weak. Proposition 3.2 only says that AA++ is better than uniform sampling in expectation. But the paper does not describe how well the points sampled by AA++ (and AA++MC) are. Also, it's unknown whether the AA++ and AA++MC are better than other non-uniform initialization in theory.
2. In figure 9 and 10, the paper shows that overall optimization time for 30 AA iterations is smaller for AA++ than for uniform initialization. However, the authors do not provide any explanation for this phnomenon.
3. The dimension of the dataset is a bit small. I suggest the authors to perform same experiments on datasets with larger dimension (d>=500).

---

> ### Author Response · Authors · 2024-03-25
>
> We thank the reviewer for the careful reading and the valuable comments.
>
> > The theoretical contribution is a bit weak. Proposition 3.2 only says that AA++ is better than uniform sampling in expectation. But the paper does not describe how well the points sampled by AA++ (and AA++MC) are. Also, it's unknown whether the AA++ and AA++MC are better than other non-uniform initialization in theory.
> > For the theory, I would like to see more elaborate analysis for AA++ and AA++MC, as that of k-means++ (Arthur & Vassilvitskii, 2007) and k-MC^2 (Bachem et al. (2016).
>
> We agree that it is unfortunate that AA++ lacks a theoretical guarantee such as the one of $k$-means++ where they bound the value of the objective function after initialization by a scaled objective function value of the optimal clustering.
> To explain why, let's first recall what $k$-means is and how it works. Lloyds algorithm consists of two steps:
>
> 1. Assignment step: Compute the distances between points and cluster centers and choosing per point the cluster center that is closest.
> 2. Update step: Per cluster, update the cluster center by computing the mean of all points assigned to this cluster.
>
> Note that those computations are rather simple and the second step can be performed in closed-form. Due to this simplicity, many theoretical analyses on $k$-means (including $k$-means++) have been published.
> In contrast to that, archetypal analysis is a harder problem, not only computationally due to the optimization problems that do not admit a closed-form solution (lines 5 and 7 in Algorithm 4), but also theoretically to analyze.
> Now, consider Lemma 3.1 in "$k$-means++: The Advantages of Careful Seeding" by Arthur & Vassilvitskii. The authors exploit (i) having closed-form solutions for the optimal cluster centers and that (ii) only points in a cluster influence the position of their cluster center. As mentioned, this is not possible for archetypal analysis since (i) there are no closed-form solutions and (ii) there is an interdependence between archetypes since we are constructing a convex hull.
>
> Furthermore, note that neither of the state-of-the-art initializations for archetypal analysis, Uniform and FurthestSum, come with any theoretical guarantees (on the objective function value similar to those of $k$-means++).
>
> We will add a paragraph after Proposition 3.2 explaining why a similar guarantee as for $k$-means++ is hard to obtain for AA++. The arguments will be similar as the explanations above.
>
> As for AA++MC, we are unsure what the reviewer means. Theorem 4.1 in the paper shows that the sampling probabilities, i.e., $p(i)$ in line 6 of Algorithm 1, of AA++MC approach those of AA++ as $m$ increases. The statement of Bachem et al. (2016) is true not only for $k$-means++ but also for AA++.

---

> > ### Comment · Reviewer_fwRK · 2024-03-26
> > **Reply**
> >
> > Thank you for your response which addresses most of my concerns. But I have one more question.
> >
> > > As for AA++MC, we are unsure what the reviewer means.
> >
> > Notice that Bachem et al. (2016) not only guarantee the convergence, but also provide a competitive ratio for the output of k-MC^2 (see thm 2 of (Bachem et al. 2016)). Is it possible to give similar theoretical results for AA++MC?

---

> > > ### Author Response · Authors · 2024-03-26
> > >
> > > Thank you for your response. We are glad that we could address most of your points.
> > >
> > > > Notice that Bachem et al. (2016) not only guarantee the convergence, but also provide a competitive ratio for the output of k-MC^2 (see thm 2 of (Bachem et al. 2016)). Is it possible to give similar theoretical results for AA++MC?
> > >
> > > Theorem 2 of Bachem et al. (2016) shows that running $k$-means++MC instead of $k$-means++ essentially yields the same bound as $k$-means++: the expected objective function value is upper bounded by a $\mathcal{O}(\log k)$ scaled objective function value of the optimal clustering, i.e., $\mathbb{E}[\phi_{k\text{-means++MC}}] \leq \mathcal{O}(\log k)\phi_\text{OPT}$. Note that within the proof of Theorem 2, Bachem et al. (2016) use the result of Arthur & Vassilvitskii (2007), i.e., $\mathbb{E}[\phi_{k\text{-means++}}] \leq \mathcal{O}(\log k)\phi_\text{OPT}$.
> > >
> > > As we argued above, we cannot (or at least we do not know how to) show such a result for AA++ and we do not believe that showing such a bound for AA++MC is easier than for AA++. It might be possible to derive a bound for AA++MC if we had one for AA++ to begin with.

---

> > > > ### Comment · Reviewer_fwRK · 2024-03-28
> > > >
> > > > Thank you for the response. I have no other questions.

---

> ### Author Response · Authors · 2024-03-25
>
> > In figure 9 and 10, the paper shows that overall optimization time for 30 AA iterations is smaller for AA++ than for uniform initialization. However, the authors do not provide any explanation for this phnomenon.
>
> The visible lines are drawn using 30 points, one per iteration of archetypal analysis. Per iteration lines 5-8 in Algorithm 4 are computed. Specifically,
>
> - line 5 involves solving $n$ non-negative least squares problems to obtain the $A$ matrix;
> - line 6 computes $Z$ given $A$ and $X$ by solving a system of linear equations;
> - line 7 involves solving $k$ non-negative least squares problems to obtain the $B$ matrix; and
> - line 8 computes $Z$ given $B$ and $X$ by computing a matrix-matrix product.
>
> There are several factors that determine the runtime of each and every iteration.
>
> 1. To solve the non-negative least squares problems, we utilize the NNLS method as stated on the bottom of page 7 and implemented in `scipy.optimize.nnls`. Note that this used to be a wrapper that calls a function implemented in FORTRAN. Only recently (we just found out ourselves as we write this rebuttal), scipy switched to a python implementation of NNLS. However, we are still using the FORTRAN implementation of NNLS and slightly adapted it: We increased the hardcoded maximum number of iterations since occasionally (very rarely) NNLS ran out of iterations. Depending on the specific inputs, the number of NNLS iterations and thus the runtime of NNLS varies.
> 2. Instead of solving a system of linear equations in line 6, we solve a least-squares problem via `np.linalg.lstsq` for cases in which $A^\top$ is not full-rank.
>
> Both factors can influence the runtime of an AA iteration.
>
> We clarify the implementation specifics in the appendix.
>
> > The dimension of the dataset is a bit small. I suggest the authors to perform same experiments on datasets with larger dimension (d>=500).
> > For the experiments, I would like to see both time comparison and performance comparison on datasets with larger dimension (d>=500).
>
> Note that we use typical benchmark data sets that were also used in related literature. Furthermore, we believe that we cover a sufficient amount and range of data set characteristics to empirically demonstrate that our proposed method and its approximations improve upon the current state of the art (Uniform and FurthestSum).
>
> Nevertheless, we added another data set to our empirical analysis: FMA (https://github.com/mdeff/fma) is a data set for music analysis that considers $n=106,574$ songs (data points) and it contains $d=518$ features (dimensions) on which we perform archetypal analysis. The results for the CenterAndMaxScale pre-processing can be seen here: https://anonymous.4open.science/r/AApp_rebuttal-4306/README.md We want to highlight that in all cases, AA++ (or one of its approximations) yields lower errors in less time, i.e., if you fix a time budget, AA++ yields a lower error than Uniform and FurthestSum.
>
> > Claims And Evidence: No
>
> May we ask the reviewer why they think that our claim(s) are not backed? Which unjustified claim(s) do we make?

---

### Decision · Action_Editor_t3U3 · 2024-04-14

**Recommendation:** Accept as is

**Comment:**

All reviewers praise the clarity and significance of the paper, and there is a clear unanimity to accept the paper. Congratulations on a fine piece of work!

**Audience:**

The paper is in a niche area (at least from a machine learning perspective) so I do not expect this to be relevant to most of the TMLR audience -- however there might be some interested readers.

**Claims And Evidence:**

All claims are supported by appropriate evidence.